# Core-sets for Fair and Diverse Data Summarization

**Sepideh Mahabadi**[*]
Microsoft Research
Redmond, WA, USA
smahabadi@microsoft.com

**Stojan Trajanovski**[*]
Microsoft
London, United Kingdom
sttrajan@microsoft.com

## Abstract

We study core-set construction algorithms for the task of Diversity Maximization under fairness/partition constraint. Given a set of points $P$ in a metric space partitioned into $m$ groups, and given $k_1, \ldots, k_m$, the goal of this problem is to pick $k_i$ points from each group $i$ such that the overall diversity of the $k = \sum_i k_i$ picked points is maximized. We consider two natural diversity measures: sum-of-pairwise distances and sum-of-nearest-neighbor distances, and show improved core-set construction algorithms with respect to these measures. More precisely, we show the first constant factor core-set w.r.t. sum-of-pairwise distances whose size is independent of the size of the dataset and the aspect ratio. Second, we show the first core-set w.r.t. the sum-of-nearest-neighbor distances. Finally, we run several experiments showing the effectiveness of our core-set approach. In particular, we apply constrained diversity maximization to summarize a set of timed messages that takes into account the messages' recency. Specifically, the summary should include more recent messages compared to older ones. This is a real task in one of the largest communication platforms, affecting the experience of hundreds of millions daily active users. By utilizing our core-set method for this task, we achieve a 100x speed-up while losing the diversity by only a few percent. Moreover, our approach allows us to improve the space usage of the algorithm in the streaming setting.

## 1 Introduction

Data summarization problem is a class of tasks where a small subset of items must be selected as a summary to represent the whole data. Typically in applications, the selected summary is required to fulfill various criteria such as *diversity*. In this paper, we focus on *Diversity Maximization* (DM) which is a topic that has attracted significant attention over the past decades [11, 33, 3, 8, 9, 10, 31, 30, 24, 19, 20, 25, 14]. The goal of this line of research is to provide a summary of a large dataset that preserves the diversity of the data as much as possible. This has many applications in various domains including summarization, recommendation systems, search, and facility location [15, 43, 40, 1, 5, 13, 2, 21, 39, 33]. Formally, in this task, given a universe $P$ of $n$ items, its goal is to choose a small subset of size $k$ that maximizes a given pre-specified measure of diversity.

*Diversity Maximization under Partition Constraints* is a variant of the problem that has been studied to find a diverse summary while satisfying some additional orthogonal constraints [7, 31, 3, 28, 4, 10, 8, 9, 6, 26]. Here, the items in the input are partitioned into $m$ disjoint groups $P = P_1 \cup \cdots \cup P_m$ and additionally, one is given a pre-specified number of desired results from each group $k_1, \ldots, k_m$, and the goal is to return $k_i$ objects from each group $P_i$ such that the overall diversity among all $k = \sum_i k_i$ points is maximized. This formulation allows to control the number of objects from each category in the output, e.g., the number of movies from each genre shown to a user in a recommendation system, or to bound the number of old messages included in a summary of a user's feed. Moreover, this

---

[*]This is an equal contribution paper.

formulation has been studied in the context of fairness (e.g. see [28, 4, 10]), where one wishes to control the number of results from each population in the produced summary. We will therefore refer to the Diversity Maximization under Partition Constraints as *Fair Diversity Maximization* (FDM), and use $\mathrm{div}_{k_1,\ldots,k_m}(P)$ to refer to the optimal achievable diversity under the fairness constraint. In this work, we consider fair diversity maximization with respect to three common diversity measures.

**Diversity Measures.** Several measures (or objective functions) have been proposed in the literature to model the notion of diversity [11, 3, 6, 25, 28, 9, 20, 16, 10, 31, 30]. A main category of such measures are based on pairwise distances (see [11] for a complete list). In this work, we will consider three of the most common pairwise distance-based measures: (1) the *minimum-pairwise* distance, where for a subset of points $S \subseteq P$, their diversity is measured as the minimum pairwise distance between the points in the subset, i.e., $\min_{p,q \in S} \mathrm{dist}(p, q)$; (2) *sum-of-pairwise* distances between the points in the subset, i.e., $\sum_{p,q \in S} \mathrm{dist}(p, q)$; and (3) the *sum-of-nearest-neighbor* distances which is an interpolation between the first two measures, and is defined formally as $\sum_{p \in S} \min_{q \in S \setminus \{p\}} \mathrm{dist}(p, q)$. We refer to these measures respectively as MIN-PAIRWISE DIST, SUM-PAIRWISE DIST, and SUM-NN DIST. All these three measures have been used previously to model diversity (e.g. see [11, 20, 6, 25], where they are respectively referred to as *remote-edge*, *remote-clique*, and *remote-pseudoforest*). In this work, we study Fair Diversity Maximization (FDM) over large datasets with respect to these three diversity measures.

**Core-sets for Diversity Maximization.** As one of the main applications of diversity maximization is related to data summarization, there has been a large body of work to solve diversity maximization in massive data models of computation, and in particular the design of *core-sets* [20, 19, 24, 27, 28, 8, 9, 41, 25, 26, 14]. Core-set is a small subset of the data which is sufficient for computing an approximate solution of a pre-specified optimization problem on the whole data.

More specifically, in this work we focus on construction of *composable core-sets*[20]: we present a summarization algorithm $\mathcal{A}$ that processes each group $P_i$ *independently* and produces a small subset of it as its summary $S_i = \mathcal{A}(P_i) \subseteq P_i$, with the property that the fair diversity of the data is approximately preserved, i.e., $\mathrm{div}_{k_1,\ldots,k_m}(S) \geq \frac{1}{\alpha} \cdot \mathrm{div}_{k_1,\ldots,k_m}(P)$, where $S$ is defined to be the union of all core-sets, i.e., $S = \bigcup_i S_i$. Composable core-sets are in particular useful for big data models of computation such as distributed and streaming settings as discussed in details in [20]. For example, in a distributed setting where the dataset is partitioned over multiple machines based on groups, each machine can compute a core-set for its own dataset, and only send this small summary over to a single aggregator. The aggregator then processes the union of the summaries and outputs the solution. We provide further applications of the core-sets for processing timed datasets in the experiments sections of our paper.

## 1.1 Prior Work

Table 1 shows a summary of prior and our results.

**Fair Diversity Maximization.** Moumoulidou *et al.* [28] studied fair diversity maximization under MIN-PAIRWISE DIST and gave an $O(m)$ approximation algorithm for the problem. The bound was later improved [4] to $m + 1$. However, it is not known whether a linear dependence on $m$ is necessary. In fact the problem admits an $O(1)$ approximation in the unconstrained version [17, 33]. For SUM-PAIRWISE DIST the problem admits a $(1/2 + \epsilon)$ approximation [3, 7], which matches the performance of the best algorithm for the unconstrained diversity maximization. Finally, for SUM-NN DIST, Bhaskara *et al.* [6] presented a randomized $O(1)$ approximation algorithm based on solving an LP for both DM and FDM.

**Core-sets for Fair Diversity Maximization.** Core-set construction algorithms for the unconstrained DM has been shown in [20] with respect to all three diversity measures. For FDM, Moumoulidou *et al.* [28] showed that running a greedy algorithm on each group independently provides a core-set with a constant approximation factor, with respect to the MIN-PAIRWISE DIST. For the SUM-PAIRWISE DIST, Ceccarello *et al.* [8] provides a core-set with the almost optimal $(1 - \epsilon)$ approximation factor. However, the size of the core-set depends on a parameter similar to the aspect ratio of the dataset. In particular, they show that for the case of doubling metrics, the core-set size could have an exponential dependence on the doubling dimension. Under SUM-NN DIST, there has been no prior work on core-sets for FDM.

Table 1: The summary of prior and our results for FDM. Here $k$ is the size of the solution and $m$ is the number of groups. All the core-set results mentioned in the table have size $poly(k)$.

|  | MIN-PAIRWISE DIST | SUM-PAIRWISE DIST | SUM-NN DIST |
|---|---|---|---|
| FDM | $O(m)$ [28, 4] | $O(1)$ [3] | $O(1)$ [6] |
| Core-set for FDM | $O(1)$ [28] | $O(1)$ **Section 2** | $O(m \cdot \log k)$ **Section 4** |

## 1.2 Our Results

**Theoretical results.** We show the following theoretical results in this paper.

- We show the first core-set construction algorithm for FDM under SUM-PAIRWISE DIST, with constant approximation factor whose size is independent of the number of points and the aspect ratio. In fact, we show a core-set of size only $k_i^2$ for each group $P_i$, that together achieve a constant factor approximation (see Theorem 2.1).

- We also show the first core-set construction algorithm for the FDM under the SUM-NN DIST notion of diversity. We provide a core-set of size $O(k^2)$ for each group $P_i$ that together achieve an $O(m \cdot \log k)$ approximation factor (see Theorem 4.1). We remark that although the approximation factor is probably not optimal, we see in the experiments section that the algorithm performs well on real data.

  To get the above core-set algorithm, we first show in Section 3, an approximation algorithm for FDM under the SUM-NN DIST. More precisely, we show an $O(m^2 \cdot \log k)$ approximation with polynomial running time (see Theorem 3.4) and $O(m \cdot \log k)$ approximation with an exponential runtime in $k$, (see Theorem 3.5). This algorithm might be of independent interest as the best previous algorithm by [6], despite having the ideal $O(1)$ approximation, is randomized and based on solving an LP, whereas our algorithm is deterministic and based on a greedy approach (see Algorithm 1). In fact all of our algorithms are simple to implement and thus practical as we show next.

- Finally, in Section C of the Appendices we show how to apply our results to the setting where the partitioning is not necessarily done according to the colors and thus get a fully composable core-set for these notions.

**Experiments.** We perform experiments for all three diversity measures. Our first set of experiments uses FDM as a tool to account for recency in a summary computed on timed datasets. Here, we group a dataset of messages by their created time. The goal is to compute a diverse summary of the messages, where we additionally require to include less number of messages in the summary from the older batches of messages, and more from the recent batches. This is a real task in one of the largest commercial communication platforms. We get the following experiments and results.

- First, we show that running diversity maximization algorithms without the group constraint does not satisfy our requirement and reports about equally number of old messages and the new ones.

- Next, we compute the price of fairness (i.e., price of *balancedness* in this context), that is we compute how much we lose diversity by imposing the grouping requirement. Our experiments on a Reddit dataset shows that the diversity only decreases by around 1% for sum-of-pairwise distances; few percent up to no more than 20% for sum-of-NN distances and around 50% for minimum-pairwise distances (due to its fragility) metrics.

- We use the core-set construction algorithms to summarize each group first, and then run our diversity maximization algorithms on the union of the core-sets. Our experiments show that using core-sets, the runtime of our algorithm improves on average by factor of $100\times$, while only losing diversity by few percent. We further remark that using core-sets in this context has an additional benefit: it removes the need to recompute the summary on the whole data when new messages arrive: once we summarize a batch of old message, we no longer need to process the batch and only need to work with the core-set that is computed once.

We further show experiments that uses FDM as a tool for controlling the desired contribution of each genre in a movie recommendation system.

Finally, we run experiments to compare our proposed core-set of Section 2 to the prior algorithm of [8] for core-set construction. We see that while the core-set produced by our algorithm is on average smaller by a factor of 200x, its performance is only worse by 1.3%.

## 1.3 Preliminaries

Throughout this work, we assume that we are given a point set $P = P_1 \cup \cdots \cup P_m$ in a metric space $(\mathcal{X}, \text{dist})$. Each point in $P$ comes from one of the $m$ groups in $[m]$, which we also refer to as *colors*. We use $P_i$ to denote the points of color $i$. We use $n$ to denote $|P|$ and $n_i$ to denote $|P_i|$. We denote the distance between two points $p, q$ by $\text{dist}(p, q)$ and for a set of points $S$, we use $\text{dist}(p, S)$ to denote $\min_{q \in S} \text{dist}(p, q)$.

### 1.3.1 Optimization problem

The optimization problem considered in this paper is Fair Diversity Maximization as defined below.

**Definition 1.1** (Fair Diversity Maximization (FDM)). Given a colored point set $P = \bigcup_i P_i$, and $k_1, \ldots, k_m$, the goal is to pick subsets $S_i \subseteq P_i$, such that first $|S_i| = k_i$, and second $\text{div}(S)$ is maximized where $S = \bigcup_i S_i$. We will use $\text{div}_{k_1,\ldots,k_m}(P)$ to denote the optimal diversity one can achieve this way, i.e.,

$$\text{div}_{k_1,\ldots,k_m}(P) = \max_{S_1 \subseteq P_1,\ldots,S_m \subseteq P_m : |S_i| = k_i} \text{div}(\bigcup_{i \leq m} S_i)$$

We will use $k$ to denote $\sum_i k_i$, and thus $|S| = k$.

**Definition 1.2** (Diversity Measures). For a set of points $S$, we consider the following diversity measures in this work : i) MIN-PAIRWISE DIST: $\min_{p,q \in S} \text{dist}(p, q)$; ii) SUM-PAIRWISE DIST: $\sum_{p,q \in S} \text{dist}(p, q)$; iii) SUM-NN DIST: $\sum_{p \in S} \min_{q \in S \setminus \{p\}} \text{dist}(p, q)$.

### 1.3.2 Summarization task

The goal in our paper is to provide an intermediate summarization (i.e., a core-set) algorithm such that the union of the core-sets contains a good solution relative to the whole data. Let us formally define this notion.

**Definition 1.3** (Core-set). Given a point set $P$, a subset $T \subseteq P$ is called an $\alpha$-approximate core-set with respect to an optimization function $f$ if the optimal value of $f$ over $T$ is within a factor $\alpha$ of the optimal value of $f$ over $P$.

It is desired that the size of the core-set is small. In this paper we focus on the optimization function $f$ being the diversity maximization function. Further, we emphasize that our algorithm for constructing core-sets for $P$, processes each group $P_i$ independent of other groups. Therefore, it provides a form of composability property as defined below.

**Definition 1.4** (Color-Abiding Composable Core-set). An algorithm $\mathcal{A}$ is said to construct an $\alpha$-approximate color-abiding composable core-set, if it produces a subset $T_i = \mathcal{A}(P_i) \subseteq P_i$ independent of the points in other colors, s.t. $\text{div}_{k_1,\ldots,k_m}(T) \geq \frac{1}{\alpha} \text{div}_{k_1,\ldots,k_m}(P)$, where $T = \bigcup_i T_i$.

Again, we note that although it is desireable that the size of the core-set $|T_i|$ is small, it does not necessarily need to be $k_i$. Later on, as a post-processing, one can use any exact or approximation algorithm on the union of the computed core-sets, i.e., $T$ to get a final solution. However the focus of this work is on the core-set computation algorithm $\mathcal{A}$.

Throughout this work, for brevity we use *composable core-set* to refer to Definition 1.4. However, we remark that the above definition of color-abiding composable core-set is only applicable when points are partitioned based on the colors, and differs from the standard notion of *composable core-sets* defined in [20], where the partitioning is done arbitrarily. Later in Section C of the Appendices, we show how to employ our algorithms and get fully composable core-sets (that is not necessarily color-abiding) as defined in [20].

### 1.3.3 The GMM algorithm

In this work we will use the greedy algorithm of [17, 33] which we denote by GMM and is depicted in Algorithm 1. Let us also state three well-known properties of GMM:

1. $r_i$'s are decreasing, i.e., for $j > i$, we have $r_i \geq r_j$.

2. For any $i < k$, let $S_i = \{p_1, \ldots, p_i\}$. Then for any $p \in P \setminus S_i$, we have that $\text{dist}(p, S_i) \leq r_{i+1}$.

3. Let $T \subset_k P$ be any subset of $k$ points in $P$. Then the minimum pairwise distance in $T$ is at most $2 \cdot r_k$.

---

**Algorithm 1** The GMM Algorithm

---

**Input** a point set $P$, and $k$
**Output** subset $S = \{p_1, \ldots, p_k\} \subseteq P$, and radii $r_1, \ldots, r_k$.
 1: $S \leftarrow$ an arbitrary point from $P$, and $r_1 \leftarrow \infty$
 2: **for** $i = 2$ **to** $k$ **do**
 3:      $p_i \leftarrow$ the farthest point in $P$ from $S$, i.e., $\text{argmax}_{p \in P} \text{dist}(p, S)$
 4:      $S \leftarrow S \cup p_i$
 5:      $r_i \leftarrow$ minimum pairwise distance in $S$, i.e., $\min_{p_1, p_2 \in S} \text{dist}(p_1, p_2)$
 6: **end for**
 7: **return** $S$

---

## 2 Core-set for FDM under SUM-PAIRWISE DIST

In this section we show an algorithm for constructing a core-set for FDM with respect to the SUM-PAIRWISE DIST notion of diversity. Given a colored point set $P = P_1 \cup \cdots \cup P_m$, and $k_1, \ldots, k_m$, the goal is to come up with a core-set construction algorithm $\mathcal{A}$ that independently summarizes each point set $P_i$ and produces a small subset of it $S_i = \mathcal{A}(P_i) \subseteq P_i$, such that $\text{div}_{k_1,\ldots,k_m}(S) \geq \frac{1}{\alpha} \cdot \text{div}_{k_1,\ldots,k_m}(P)$, where again $S = \bigcup_i S_i$.

**Overview of the algorithm.** Suppose that we want to find a solution that maximizes FDM under SUM-PAIRWISE DIST notion of diversity. Consider the optimal solution OPT and let $\text{OPT}_i$ be the set of points from color $i$. Our algorithm for constructing a core-set proceeds by running GMM for $k_i$ iterations on $P_i$, to get a *seed* of size $k_i$. It is then a property of GMM that if each point $p \in \text{OPT}_i$ is mapped (i.e. moved) to the closest seed in $P_i$, its distance does not change by more than $r_i$ (as defined in GMM). Now the points in different colors can have different $r_i$ and therefore the movements can be of different scales. For two points $p, q \in \text{OPT}$, if the scales of their movements w.r.t. their original distance are small, then the new distance is still large compare to before. Otherwise, if the scale of these movements are large, we show how to charge the loss in the diversity of this pair onto other pairs that have relatively small movements. If there are not enough such pairs, one can then show that the solution returned by the seeds themselves have had large diversity. Finally, in order to make the map injective (so that no two points are mapped to the same core-set point), for each of the seed points, the algorithm also stores in the core-set, the closest $k_i$ points to the seed, and thus increasing the size of the core-set to $k_i^2$.

There is one hard case in the above approach and that is when $k_i = 1$. In this case, the GMM algorithm can return any arbitrary point as the seed and that could provably be a bad core-set. To handle this case, we show that it is enough for the core-set algorithm to find at least a minimum of two seeds. Then we show that it is always possible to pick one of the seeds and get a reasonably good approximation. This modification is proved to work in Appendix B.

**Algorithm description.** For simplicity, let us assume that for all $i$, we have $k_i \geq 2$. Otherwise, we show in Appendix B that a slight modification of the algorithm works (although the proof becomes more involved). The core-set construction algorithm is shown in Algorithm 2. The algorithm proceeds by running GMM for $k_i$ iterations to compute a set of $k_i$ centers. Then for each of these $k_i$ centers such as $p$, the algorithm stores at most $k_i$ points from $P_i$ whose closest center among all $k_i$ centers is $p$.

**Theorem 2.1.** *Algorithm 2 produces a core-set with size $O(k_i^2)$ and a constant factor approximation:* $\text{div}_{k_1,\ldots,k_m}(S) \geq \frac{1}{C} \cdot \text{div}_{k_1,\ldots,k_m}(P)$ *for a constant $C$. (Proof in Appendix A.1)*

---

**Algorithm 2** Core-set Construction Algorithm for SUM-PAIRWISE

---

**Input** a point set $P_i$, together with parameters $k_i$ and $k$ (where $k = k_1 + \cdots + k_m$)
**Output** a subset $S_i \subseteq P_i$
1:  $S_i = \{p_1, \ldots, p_{k_i}\} \leftarrow \text{GMM}(P_i, k_i)$
2:  $T \leftarrow \emptyset$
3:  **for** $p \in S_i$ **do**
4:      **for** $j = 1$ **to** $k_i$ **do**
5:          $T \leftarrow T \cup$ any point $p_j \in P_i \setminus T$ s.t. $\text{argmin}_{q \in S_i} \text{dist}(p_j, q) = p$.
6:      **end for**
7:  **end for**
8:  $S_i \leftarrow S_i \cup T$
9:  **return** $S_i$

---

## 3   Approximation Algorithm for FDM under SUM-NN DIST

As mentioned in the results section, in order to obtain our core-set construction algorithm under SUM-NN DIST in Section 4, first in this section we show an approximation algorithm for FDM under the SUM-NN DIST notion of diversity.

**Overview of the algorithm.** Consider the optimal solution OPT that maximizes FDM under SUM-NN DIST notion of diversity, and let $\text{OPT}_i$ be the set of points from color $i$. Now each point $p \in \text{OPT}$ has a contribution towards the diversity, i.e., $\text{dist}(p, \text{OPT} \setminus \{p\})$, and thus the contribution of each group $\text{OPT}_i$ towards the optimal solution is well-defined. Let us call the color with the maximum contribution as the *most significant* color. Clearly if we find a solution SOL whose value is at least as large as the contribution of the most significant color, then we are within a factor of $m$ of the optimal solution.

Next, suppose that the most significant color is $i$ and let $d_1 \geq \cdots \geq d_{k_i}$ be the contributions of the points in the optimal solution that are of color $i$. It is not hard to prove (as we will show later) that $\max_j (j \cdot d_j)$ approximates $\sum_{j=1}^{k_i} d_j$ up to a factor of $\log k$. So we will construct a solution whose value is at least $j \cdot d_j$ and this will be within a factor of $(m \cdot \log k)$ of the optimal solution as promised.

More concretely, we will construct a solution such that it contains $j$ points of color $i$ where no other point (from any color) lies within a distance $d_j$ of them (in reality, we find $O(j)$ points such that no other point lies within a distance $O(d_j)$ of them). Of course, we do not know $i$, $j$, and $d_j$ *a priori*. We can enumerate $i$ and $j$. Further, if we run the GMM algorithm on color $i$ for $j$ iterations, then the minimum pairwise distance between the retrieved points is an approximation to $d_j$.

This itself is enough to pick $O(j)$ points of color $i$ that are $O(d_j)$ far from each other. Let us call these points the seeds. However, the remaining challenge is to pick the rest of the points (that is the $k_i - O(j)$ points of color $i$, as well as $k_\ell$ points of color $\ell$ for all $\ell \neq i$) away from these seeds.

This brings us to the following problem: given a colored point set $P$ and a set of at most $k$ balls $B$, how to pick the largest subset of balls $B' \subseteq B$, such that enough points exist *outside* of $B'$. (Note that $B$ is basically the set of balls with radius $O(d_j)$, that are centered at the $j$ points returned by the first $j$ iterations of GMM). This problem can be solved exactly by an exhaustive search and spending exponential in $k$ time. One can also get a polynomial time $O(m)$-approximate solution for the problem by iteratively excluding half of the remaining balls from the solution, and at the same time satisfying the condition for half of the colors. Thus, after $O(\log m)$ iterations, all color constraints are satisfied, and a $1/m$ fraction of the balls remain. There are further technical details needed for the proof to go through which we need to take care of.

### 3.1   Algorithm Description

Given a colored point set $P = P_1 \cup \cdots \cup P_m$, and $k_1, \ldots, k_m$, the goal is to find a solution $\text{SOL} = \text{SOL}_1 \cup \cdots \cup \text{SOL}_m$ where $\text{SOL}_i \subseteq P_i$, and $|\text{SOL}_i| = k_i$, and that $\text{div}(\text{SOL}) \geq \frac{1}{\alpha} \cdot \text{div}_{k_1, \ldots, k_m}(P)$. The approximation algorithm is shown in Algorithm 3. The only unspecified part of the algorithm is in Line 8, where we need to specify how we find the subset $B'$ given the set of balls $B$. One can take different approaches resulting in different trade-offs described below.

**Algorithm 3** Approximation Algorithm for FDM under SUM-NN DIST Notion of Diversity

---

**Input** a colored point set $P = P_1 \cup \cdots \cup P_m$, and $k_1, \ldots, k_m$
**Output** a solution SOL $\subseteq P$ where $|\text{SOL} \cap P_i| = k_i$, with approximately maximum diversity

1: SOL $\leftarrow$ an arbitrary solution satisfying the color constraints, i.e., $|\text{SOL} \cap P_i| = k_i$
2: **for** $i \in [m]$ **do**
3:    $G_i = \{p_1, \ldots, p_k\} \leftarrow \text{GMM}(P_i, k)$, and let $r_1, \ldots, r_k$ be their corresponding radii.
4:    **for** $j = 2$ **to** $k$ **do**
5:       $S_1, \ldots, S_m \leftarrow \emptyset$
6:       Let $t(j) \geq j$ be the largest iteration where $r_{t(j)} \geq r_j/2$ and let $t(j) = k$ if no such iteration exists.
7:       Let $B = \{B_1, \ldots, B_{t(j)}\}$ be the balls of radius $r_{t(j)}/2$ around the points $p_1, \ldots, p_{t(j)}$
8:       *Find* an (approximately) largest subset of balls $B' \subseteq B$ such that $P \setminus B'$ contains at least $k_i - |B'|$ points from color $i$, and at least $k_\ell$ points from color $\ell$ for each color $\ell \neq i$.
9:       Add centers of $B'$ to $S_i$
10:      Add an arbitrary set of $k_i - |B'|$ points from $P_i \setminus B'$ to $S_i$
11:      **for** $\ell \neq i$ **do**
12:         Add an arbitrary set of $k_\ell$ points from $P_\ell \setminus B'$ to $S_\ell$
13:      **end for**
14:      **if** $S = \bigcup_i S_i$ is a valid solution **and** $\text{div}(S) > \text{div}(\text{SOL})$ **then**
15:         SOL $\leftarrow S$
16:      **end if**
17:    **end for**
18: **end for**
19: **return** SOL

---

### 3.1.1 Finding an optimal subset of balls (Line 8 of Algorithm 3)

Given a colored point set $P$ and a set of disjoint balls $B$, the goal here is to find an (approximately) largest subset of balls $B' \subseteq B$, such that

- For a pre-specified color $i \in [m]$, we have $|P_i \setminus B'| \geq k_i - |B'|$,
- For all other colors $\ell \in [m] \setminus \{i\}$, we have $|P_\ell \setminus B'| \geq k_\ell$.

Moreover, we may assume that $|B| \leq t(j) \leq k$, and that we are interested only in subsets $B'$ of size upto $|B'| \leq k_i$.

**Approach 1.** If the algorithm is allowed to spend exponential time in $k$, one can find the optimal subset $B'$ in time $k^{k_i} \cdot m$ as stated below.

**Observation 3.1** (Approach 1). *There is an algorithm that finds the optimal largest subset $B' \subseteq B$ in time $\binom{t_j}{j} \cdot m \cdot j = k^{O(k_i)}$. (Proof in Appendix A.2).*

**Approach 2.** One can get an approximate solution in polynomial time if the number of points of each color is sufficiently large, i.e, for each $\ell \in [m]$, we have $n_\ell = |P_\ell| \geq 2 \cdot k_\ell$. Note that this is a realistic assumption (also appeared in [11]), as we expect the number of data points to be much larger than the target size for the summary. Further, this assumption can be relaxed to $n_\ell$ being larger than a constant times $k_\ell$ similar to [11].

**Lemma 3.2.** *There is an algorithm that finds an $O(m)$-approximate largest subset $B' \subseteq B$ in polynomial time, under the assumption that for each $\ell \in [m]$, we have $n_\ell \geq 2 \cdot k_\ell$. (Proof in Appendix A.3).*

### 3.2 Algorithm Analysis

Let us now describe the intuition behind Algorithm 3 along with defining a set of notations. Let $\text{OPT} \subseteq P$ be the optimal solution and let $\text{OPT}_i = \text{OPT} \cap P_i$ be the points of color $i$ in the optimal solution. Moreover let $d_1^i, \ldots, d_{k_i}^i$ be the values of the optimal solution, i.e., $d_j^i = \text{dist}(p_j^i, \text{OPT} \setminus \{p_j^i\})$, where $p_j^i$'s are the points in $\text{OPT}_i$. Therefore, we have that $\text{div}_{k_1, \ldots, k_m}(P) = \text{div}(\text{OPT}) = \sum_{i \leq m} \sum_{j \leq k_i} d_j^i$.

Let $i^* \leq m$ be the color with the maximum contribution in the optimal solution, i.e., $i^* = \operatorname{argmax}_{i \in [m]} \sum_{j \leq k_i} d_j^i$. Clearly, $\sum_{j \leq k_{i^*}} d_j^{i^*} \geq \operatorname{div}(\text{OPT})/m$. Moreover, WLOG assume that for each $i$, $d_j^i$'s are sorted, i.e., $d_1^i \geq \cdots \geq d_{k_i}^i$. Now, let $j^*(i)$ be defined as $j^*(i) = \operatorname{argmax}_{j \leq k_i} j \cdot d_j^i$, and define $j^* = j^*(i^*)$.

**Claim 3.3.** *Let $i \in [m]$ be any color. Then we have $j^*(i) \cdot d_{j^*(i)}^i \geq \frac{1}{\lceil \log(k_i+1) \rceil} \sum_{j \leq k_i} d_j^i$. (Proof in Appendix A.4)*

**Theorem 3.4.** *Algorithm 3 using Approach 2, runs in polynomial time and produces a solution with an approximation factor of $O(m^2 \cdot \log k)$. (Proof in Appendix A.5)*

Similarly, running Algorithm 3 with Approach 1 would give us the following theorem.

**Theorem 3.5.** *There exists an $O(m \cdot \log k)$ approximation algorithm that runs in time $O(k^k \cdot poly(m, k))$.*

# 4 Core-set for FDM under SUM-NN DIST

In this section we show how to get a composable core-set for FDM with respect to the SUM-NN DIST notion of diversity.

**Overview of the algorithm.** In order to construct a core-set for FDM under the SUM-NN DIST notion of diversity, we show that it is enough to construct a subset of the points such that the solution to the problem of finding the maximum number of balls such that enough points exist outside of them (discussed in Section 3) remains unchanged. We show that the GMM algorithm has this property. Intuitively, since GMM chooses the points that are pairwise-far from each other, not too many points are picked inside individual balls and thus the solution to the balls problem remains relatively unchanged. Therefore, if we run the algorithm of Section 3 on the core-sets produced by GMM, the solution is similar as if we had run it on the whole data.

**Algorithm description.** In this section, the goal is to give a summarization algorithm $\mathcal{A}$ that processes each dataset $P_i$ independently of other color datasets $P_j$ and produces a summary $S_i = \mathcal{A}(P_i)$ such that $\operatorname{div}_{k_1,\ldots,k_m}(S) \geq \frac{1}{\alpha} \cdot \operatorname{div}_{k_1,\ldots,k_m}(P)$ where again $S = \bigcup_i S_i$. The core-set construction algorithm is simple and is shown in Algorithm 4. The algorithm proceeds by running the GMM algorithm $k$ times, and each time for $k + 1$ iterations, and without replacement.

---

**Algorithm 4** Core-set Construction Algorithm for SUM-NN

---

**Input** a point set $P_i$, together with parameters $k_i$ and $k$ (where $k = k_1 + \cdots + k_m$)
**Output** a subset $S_i \subseteq P_i$

1: $S_i \leftarrow \emptyset$
2: **for** $j = 1$ **to** $k$ **do**
3: $\quad G_i = \{p_1, \ldots, p_{k+1}\} \leftarrow \text{GMM}(P_i, k+1)$
4: $\quad S_i \leftarrow S_i \cup G_i$
5: $\quad P_i \leftarrow P_i \setminus G_i$
6: **end for**
7: **return** $S_i$

---

**Theorem 4.1.** *Algorithm 4 returns a composable core-set of size $O(k^2)$ for each of the $m$ colors, with an approximation factor of $O(m \cdot \log k)$. (Proof in Appendix A.6)*

# 5 Experiments

To demonstrate the effectiveness of our algorithms, we run simulations on public and timed datasets.

## 5.1 Tasks and Datasets

**FDM as a tool to account for recency in a summary.** Here the task is to produce a summary of a set of timed messages with a property that (i) the summary is a diverse subset of the messages, and (ii) there are more recent messages shown in the summary. We model this task by partitioning the messages into groups / colors based on their creation times and assign a desired budget to each group.

We use a *Reddit* dataset [42] of text messages that are semantically embedded using BERT [12] into a metric space. We also utilize message creation time stamps as we aim to show diverse, yet timely and relevant messages to a user created across different time-window intervals within a certain period and thus assign a color to each message based on to which time-window interval (e.g., a week, thus having 4 colors in total) the message creation time belongs to.

**FDM as a tool for controlling the desired fairness contribution of each group.** We further use FDM to control the contribution of each genre (besides time) in a movie dataset. Apart from the *Reddit* dataset, for this task we also use the *MovieLens* dataset [18] where the movie titles are semantically embedded into a metric space. For this dataset, we have assigned a movie to a group represented by a color based on two criteria: (i) movie genre, and (ii) the time-window interval the movie review creation time belongs to. We include the result of this dataset in Appendix D.2.

More details on the datasets preparation are given in Appendix D.1 and additional result figures are provided in Appendix D.2. We have also made the code of the algorithms publicly available[2].

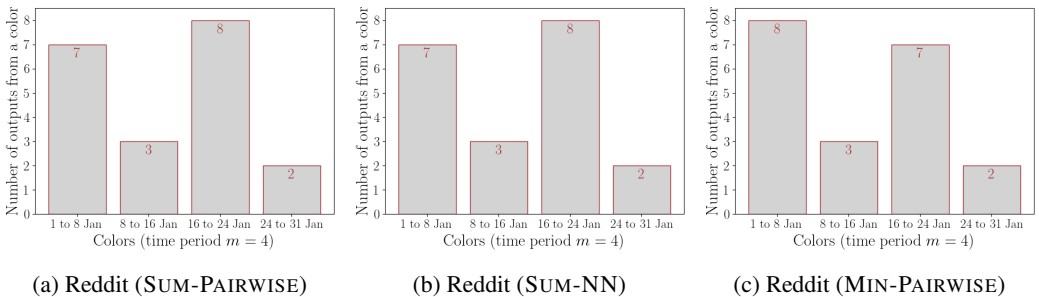

(a) Reddit (SUM-PAIRWISE)    (b) Reddit (SUM-NN)    (c) Reddit (MIN-PAIRWISE)

Figure 1: DM algorithm outcomes for all colors ($m = 4$) with equidistant time period as fairness colors in the Reddit dataset ($k = 20$ items).

## 5.2    Summary of the experiments and results

We run the following experiments using all three diversity measures.

**Need for FDM.** We first show why we need to resort to FDM. In particular, we show that if we run DM on the data, the results are not as fairly balanced as we want, as depicted in Figure 1 and Figures 2 & 3 (Appendix D.2.1). In the case of time periods as colors dividing the data into $m = 4$ colors (i.e., quarters within a month), for $k = 20$ DM algorithms give: (i) a certain color that is clearly dominantly present in the outcome (for all diversity distances), and (ii) this color is not the one from the most recent messages (i.e., from the last color), see Figure 1 for the *Reddit* dataset.

**Price of fairness (balancedness).** Our next set of experiments show that when using FDM the diversity is not decreased by a lot compared to DM. We use the *loss* of diversity (% Div. loss) expressed as a relative change of diversity distances as a measure. This is shown in Table 2, and Tables 4 & 5 (Appendix D.2.2). For most of these experiments we use the state-of-the-art algorithms. However for SUM-NN DIST we use our proposed deterministic algorithm of Section 3 as an alternative to the LP-based randomized algorithm of [6]. When we run FDM, we gain in fairness (or balancedness), but we lose in diversity over the DM, expressed in different metrics. Experiments from Table 2 show that we lose around 1% for SUM-PAIRWISE DIST; from few percent up to no more than 20% for SUM-NN DIST and around 50% for MIN-PAIRWISE DIST (due to its fragility) for various color distributions per group, while achieving the desired per group distribution, in both cases where we enforce balancedness (i.e., uniform $k_i$), and when we enforce recency (i.e., $k_i$ increasing). The results also show a similar trend if we use alternative message embeddings (Table 7 in Appendix D.2.4).

**Effectiveness of our core-sets.** We then show the effectiveness of the core-sets. We run our core-set construction algorithms (of Section 2 and Section 4) on each color independently, to get a smaller size dataset. We then run the FDM optimization once on the union of the core-sets and once on the whole data. We also measure the *loss* of diversity and runtime improvement achieved by the use of core-sets. The results for all three measures are given in Table 3 and Table 6 (Appendix D.2.3). In

---

[2]`https://github.com/microsoft/coresets-fair-diverse`

Table 2: The *loss* of diversity (% Div. loss) between DM vs. FDM on the full data, expressed as a relative change, of the concerned SUM-PAIRWISE, SUM-NN and MIN-PAIRWISE distances for uniform (upper part) or increasing (lower part) color values $k_i$ for the Reddit dataset.

| DM vs. FDM | | SUM-PAIRWISE | SUM-NN | MIN-PAIRWISE |
|---|---|---|---|---|
| colors $k_i$ | $\sum k_i$ | % Div. loss | % Div. loss | % Div. loss |
| $[2, 2, 2, 2]$ | 8 | 1.22% | 9.66% | 51.57% |
| $[3, 3, 3, 3]$ | 12 | 0.98% | 14.27% | 49.99% |
| $[4, 4, 4, 4]$ | 16 | 0.50% | 13.72% | 48.78% |
| $[5, 5, 5, 5]$ | 20 | 0.47% | 18.96% | 48.05% |
| $[6, 6, 6, 6]$ | 24 | 0.19% | 9.48% | 47.20% |
| $[2, 4, 6, 8]$ | 20 | 0.42% | 15.40% | 48.05% |
| $[3, 6, 9, 12]$ | 30 | 0.29% | 13.29% | 46.34% |
| $[4, 8, 12, 16]$ | 40 | 0.25% | 1.98% | 45.52% |
| $[5, 10, 15, 20]$ | 50 | 0.16% | 9.62% | 44.48% |
| $[6, 12, 18, 24]$ | 60 | 0.12% | 3.98% | 43.60% |

terms of diversity loss, the results show the diversity values are on-par or with marginal difference if we apply the FDM to the union of core-sets, compared to FDM applied to the full data. (In the case of SUM-NN DIST, there are cases where one is marginally better than the other, due to higher approximation factor.) Our experiments show that using core-sets, the runtime of our algorithm improves on average by factor of few $10\times$ or $100\times$, while only losing diversity by few percent. We remark that using core-sets in this context has an additional benefit: it removes the need to recompute the summary on the whole data when new messages arrive: once we summarize a batch of old message, we no longer need to process the batch and only need to work with the core-set that is computed once.

Table 3: The *loss* of diversity (% Div. loss) expressed as a relative change of diversity distances, and the running time *gains* ($\times$ times faster) of the FDM when applied to the union of core-sets compared to FDM applied to the full data for uniform or increasing color values $k_i$ for the Reddit dataset. We remark that the results for MIN-PAIRWISE have % Div. loss being 0% as for this data the two points connected with the minimum distance always end up in the union of the core-sets.

| FDM full data vs. core-sets | | SUM-PAIRWISE | | SUM-NN | | MIN-PAIRWISE | |
|---|---|---|---|---|---|---|---|
| colors $k_i$ | $\sum k_i$ | % Div. loss | Time gain ($\times$) | % Div. loss | Time gain ($\times$) | % Div. loss | Time gain ($\times$) |
| $[2, 2, 2, 2]$ | 8 | 1.35% | 196.24 | 2.22% | 1 769.70 | 0.00% | 208.64 |
| $[3, 3, 3, 3]$ | 12 | 0.67% | 333.13 | 0.29% | 888.55 | 0.00% | 152.48 |
| $[4, 4, 4, 4]$ | 16 | 1.21% | 539.69 | −1.59% | 474.26 | 0.00% | 122.29 |
| $[5, 5, 5, 5]$ | 20 | 1.17% | 432.68 | −0.44% | 294.23 | 0.00% | 89.08 |
| $[6, 6, 6, 6]$ | 24 | 0.94% | 130.87 | −3.03% | 183.28 | 0.00% | 63.69 |
| $[2, 4, 6, 8]$ | 20 | 1.50% | 845.98 | −1.80% | 285.68 | 0.00% | 91.44 |
| $[3, 6, 9, 12]$ | 30 | 1.06% | 134.76 | 2.27% | 110.36 | 0.00% | 53.05 |
| $[4, 8, 12, 16]$ | 40 | 1.02% | 182.06 | −0.88% | 57.88 | 0.00% | 36.51 |
| $[5, 10, 15, 20]$ | 50 | 1.16% | 194.36 | 0.71% | 34.90 | 0.00% | 26.97 |
| $[6, 12, 18, 24]$ | 60 | 1.27% | 172.25 | −0.49% | 23.71 | 0.00% | 20.53 |

**Comparison to state-of-the-art core-set algorithm for SUM-PAIRWISE DIST.** Finally, we compare the core-set construction algorithm in [8] for SUM-PAIRWISE DIST to our algorithm of Section 2. The experiments in Appendix D.2.5 show that, the size of the core-set obtained by the algorithm in [8] for getting a factor 2 approximation is close to the data size in practice. In addition, the final diversity loss of our algorithm is negligible (less than 1% to few percent) even though it produces a much smaller core-set. Further, our core-set construction algorithm is also significantly faster.

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

# A  Deferred Proofs

## A.1  Proof of Theorem 2.1

*Proof.* First note that it is easy to see that the size of each $S_i$ is at most $k_i^2$.

We will then show that the approximation factor is $O(1)$. Let $\text{OPT}_i = \text{OPT} \cap P_i$ be the points of color $i$ in the optimal solution of $P$. For each color $i \in [m]$, let us use $r_i$ to denote the minimum pairwise distance between the set of $k_i$ points returned by $\text{GMM}(P_i, k_i)$ in Line 1 of the algorithm. Further let us abuse the notation, and for a point $x \in P$, use $r_x$ to denote $r_{c_x}$ where $c_x \in [m]$ is the color of the point $x$.

Now let us also divide the optimal value of the diversity into two parts.

$$\text{div}(\text{OPT}) = \sum_{x,y \in \text{OPT}:\ \text{dist}(x,y) < 5\max\{r_x, r_y\}} \text{dist}(x, y) + \sum_{x,y \in \text{OPT}:\ \text{dist}(x,y) \geq 5\max\{r_x, r_y\}} \text{dist}(x, y)$$

Let $A := \sum_{x,y \in \text{OPT}:\ \text{dist}(x,y) < 5\max\{r_x, r_y\}} \text{dist}(x, y)$ be the first term in the above summation and $B := \sum_{x,y \in \text{OPT}:\ \text{dist}(x,y) \geq 5\max\{r_x, r_y\}} \text{dist}(x, y)$ denote the second term. Now we consider two cases separately.

**Case 1: $A \geq B$.**  In this case we have that

$$\text{div}(\text{OPT}) \leq 2 \cdot A \leq 2 \sum_{x,y \in \text{OPT}} 5 \cdot \max\{r_x, r_y\} \leq 20 \sum_{x,y \in \text{OPT}} r_x \leq 20 \cdot k \sum_{i \in [m]} k_i r_i$$

Now let us define a solution as follows. Let $\text{SOL}_i \subseteq S_i$ be the first set of $k_i$ points added to $S_i$ in Line 1 of the algorithm using GMM. Then clearly the overall diversity of this solution is at least

$$\text{div}(\text{SOL}) = \sum_{i \in [m]} \sum_{x,y \in \text{SOL}_i} \text{dist}(x, y) + \frac{1}{2} \cdot \sum_{i \in [m]} \sum_{y \in \text{SOL} \setminus \text{SOL}_i} \sum_{x \in \text{SOL}_i} \text{dist}(x, y)$$

$$\geq \sum_{i \in [m]} \binom{k_i}{2} r_i + \frac{1}{2} \cdot \sum_{i \in [m]} \sum_{y \in \text{SOL} \setminus \text{SOL}_i} \frac{\binom{k_i}{2}}{(k_i - 1)} r_i \qquad \text{by the triangle inequality}$$

$$\geq \sum_{i \in [m]} r_i \left( \binom{k_i}{2} + (k - k_i) \cdot \frac{k_i}{4} \right)$$

$$\geq \frac{k}{4} \sum_{i \in [m]} k_i r_i + \sum_{i \in [m]} r_i \left( \frac{k_i^2}{4} - \frac{k_i}{2} \right)$$

$$\geq \frac{k}{4} \sum_{i \in [m]} k_i r_i$$

where to get the last inequality, we used the fact that for $k_i > 1$ the second term is non-negative. Therefore $\text{div}(\text{SOL}) \geq \frac{1}{80} \cdot \text{div}(\text{OPT})$.

**Case 2: $A < B$.**  In this case we have that $\text{div}(\text{OPT}) \leq 2B$. We will show how to choose the solution SOL. Let $\text{OPT}_i = \text{OPT} \cap P_i$ as before.

**Observation A.1.** *There exists a one-to-one mapping $\mu : \text{OPT}_i \to S_i$ s.t. for each $o \in \text{OPT}_i$, $\text{dist}(o, \mu(o)) \leq 2r_i$.*

*Proof.* Let $S_i' \subseteq S_i$ be the first set of $k_i$ points added to $S_i$ in Line 1 of the algorithm. For a point $p \in P_i$ let $n(p) = \operatorname{argmin}_{s' \in S_i'} \text{dist}(p, s')$ be the nearest center in $S_i'$ to $p$. Also, for a center $s' \in S_i'$, let $N(s') = \{s \in S_i : n(s) = s'\}$ be the set of points in the core-set whose nearest center in $S_i'$ is $s'$, i.e., the points that have been added to the core-set in Line 5 of the algorithm while processing $s'$. Finally let $D(s') = \{o \in \text{OPT}_i : n(o) = s'\}$ be the set of points of color $i$ in the optimal solution whose closest center in $S'$ is $s'$. Note that because $|\text{OPT}_i| = k_i$, we have $|N(s')| \geq |D(s')|$, as the algorithm keeps adding points to $S_i$ as long as such point exists, and fewer than $k_i$ points

have been added to $N(s')$. Therefore, we can always find a matching between $D(s')$ and $N(s')$ of size at least $|D(s')|$. Thus such a $\mu$ exists that maps a point $o \in D(s')$ to its match in $N(s')$. By Property 3 of GMM we know that $\mathrm{dist}(o, s') \leq r_i$ and $\mathrm{dist}(s', \mu(o)) \leq r_i$, and thus we get that $\mathrm{dist}(o, \mu(o)) \leq 2r_i$. $\qquad\square$

Now define $\mathrm{SOL}_i \subseteq S_i$ to be $\{\mu(o) \colon o \in \mathrm{OPT}_i\}$ and clearly $|\mathrm{SOL}_i| = k_i$. Then we have

$$
\begin{aligned}
\mathrm{div}(\mathrm{SOL}) \geq & \sum_{x,y \in \mathrm{OPT}\,\colon\, \mathrm{dist}(x,y) \geq 5\max\{r_x, r_y\}} \mathrm{dist}(\mu(x), \mu(y)) \\
\geq & \sum_{x,y \in \mathrm{OPT}\,\colon\, \mathrm{dist}(x,y) \geq 5\max\{r_x, r_y\}} \mathrm{dist}(x, y) - \mathrm{dist}(x, \mu(x)) - dist(y, \mu(y)) \\
\geq & \sum_{x,y \in \mathrm{OPT}\,\colon\, \mathrm{dist}(x,y) \geq 5\max\{r_x, r_y\}} \mathrm{dist}(x, y) - 2r_x - 2r_y \\
\geq & \sum_{x,y \in \mathrm{OPT}\,\colon\, \mathrm{dist}(x,y) \geq 5\max\{r_x, r_y\}} \mathrm{dist}(x, y)/5 \\
\geq & \; B/5 \geq \mathrm{div}(\mathrm{OPT})/10
\end{aligned}
$$

$\qquad\square$

## A.2   Proof of Observation 3.1

*Proof.* One can brute force on each possible subset $B'$, and count the number of points from each color that falls in $B'$, and thus compute the number of remaining points for each color, i.e., $|P_\ell \setminus B'|$ in $O(|B'| \cdot m)$ time and check whether the number exceeds $k_\ell$. So the total runtime is at most $\sum_{z=1}^{k_i} \left( \binom{t(j)}{z} \cdot z \cdot m \right) \leq m \cdot k^{k_i}$. $\qquad\square$

## A.3   Proof of Lemma 3.2

*Proof.* First one can check if there exists a subset $B'_0$ of size $2 = O(1)$ in polynomial time using Approach 1.

Next we construct a solution $B_1$. For each $\ell \in [m]$, let $d_\ell = k_\ell$ be the demand of color $\ell$, that is the total number of points of that color which we want to exist outside of our solution $B_1$. First WLOG we assume that all points of $P$ lie inside one of the balls in $B$ (if this is not the case, one can subtract the number of points of each color that are outside of all the balls from the demands $d_\ell$). Then we start with $B_1 = B$ and repeat the following process as long as there are non-zero demands left.

- We let $A$ be an arbitrary subset of half of the balls in $B_1$ and let $\bar{A} = B_1 \setminus A$.

- Note that for each color $\ell \in [m]$ with non-zero demand $d_\ell > 0$, either $A$ or $\bar{A}$ contains at least $d_\ell$ points of color $\ell$ in them. This is initially true as one of $A$ or $\bar{A}$ should contain half of the points that is $n_\ell/2 \geq k_\ell \geq d_\ell$. We show this remains true at the end of the process.

- Let $t_A$ be the number of colors $\ell \in [m]$ with non-zero demands (i.e., $d_\ell > 0$) such that $A$ contains more points of color $\ell$ than $\bar{A}$, and define $t_{\bar{A}}$ similarly as the number of colors $\ell \in [m]$ with non-zero demands such that $\bar{A}$ contains more points of color $\ell$ than $A$.

- Now we let $B_1 = A$ if $t_{\bar{A}} \geq t_A$ and let $B_1 = \bar{A}$ otherwise. This ensures that the demand of at least half of the colors with non-zero demand has been satisfied as the set we are excluding from $B_1$ satisfies the demand of at least half of the colors with non-zero demands. This means that the total number of iterations is at most $\log m$.

- We will update the demands of all the colors based on the set that was excluded from $B_1$. WLOG assume that $t_A \geq t_{\bar{A}}$. Then for each $\ell$ that $d_\ell > 0$ we set $d_\ell = \max\{0, d_\ell - |A \cap P_\ell|\}$.

  Let $\ell$ be a color that has still a non-zero demand. Note that this means that $|A \cap P_\ell| \leq d_\ell$. Therefore we will still have the property that $|\bar{A} \cap P_\ell| = |(B_1 \setminus A) \cap P_\ell| \geq 2d_\ell -$

$|A \cap P_\ell| \geq 2(d_\ell - |A \cap P_\ell|)$, where here we used $B_1$ and $d_\ell$ to denote their values before we apply the update corresponding to this iteration.

Finally, we return the best of $B_0$ and $B_1$ as $B'$, i.e., the one with maximum size. Note that if $\text{OPT} \leq 2m$, then $B_0$ itself is an $O(m)$ approximation. Otherwise, if $\text{OPT} > 2m$, then as the total number of iterations to obtain $B_1$ is at most $\log m$, we have $|B'| = |B_1| \geq |B|/2^{\log m} = |B|/m \geq \text{OPT}/m$. $\qquad\square$

## A.4   Proof of Claim 3.3

*Proof.* Divide the interval $[1, k_i]$ into logarithmically many number of intervals such that for each $1 \leq \ell \leq \lceil \log(k_i + 1) \rceil$, $I_\ell = [2^{\ell-1}, \min\{2^\ell, k_i + 1\})$. Then for each $\ell$, we have that

$$\sum_{j \in I_\ell} d_j^i \leq 2^{\ell-1} \cdot d_{2^{\ell-1}}^i \leq j^*(i) \cdot d_{j^*(i)}^i$$

and therefore, $\sum_{j \leq k_i} d_j^i \leq \lceil \log(k_i + 1) \rceil \cdot j^*(i) \cdot d_{j^*(i)}^i$. $\qquad\square$

## A.5   Proof of Theorem 3.4

*Proof.* First note that, it can be easily checked that the algorithm runs in polynomial time by Lemma 3.2. Next, consider the iteration of Algorithm 3 corresponding to $i$ and $j$ where $i = i^*$ and $j$ is chosen as follows,

1. $j$ is the largest value in the range $j^* \leq j \leq k$ such that $r_j \geq d_{j^*}^{i^*}$

2. If no such $j$ exists, we let $j = j^*$.

3. Finally, if $j^* < j < k$ and $r_{j+1} \geq d_{j^*}^{i^*}/2$, increase $j$ by one, i.e., $j = j + 1$.

In all of the above cases, either by Property 3 of the GMM algorithm, or by definition, we have that $r_j \geq d_{j^*}^{i^*}/2$. Therefore, $r_{t(j)} \geq d_{j^*}^{i^*}/4$. Therefore the balls in $B$ defined in Line 7 of the algorithm, have radius at least $d_{j^*}^{i^*}/8$.

**Claim A.2.** *Let $B'' \subseteq B$ be the maximum size subset of the balls s.t. $P \setminus B''$ contains $k_i - |B''|$ points of color $i$, and at least $k_\ell$ number of points from color $\ell$, for each $\ell \neq i$. Then $|B''| \geq j^*$.*

*Proof.* Note that $\text{OPT}_{i^*}$ contains a subset $R$ of $j^*$ points from $P_{i^*}$ such that there are no other points from OPT in the ball of radius $d_{j^*}^{i^*}$ around the points in $R$. Now, we consider three cases separately.

**Case 1:** $j = k$. Note that in this case we also have that $t(j) = k$. Therefore, $B$ contains $k$ disjoint balls. Obtain $B''$ from $B$ as follows. First set $B'' = B$. Then for each color $\ell \neq i$, remove at most $k_\ell$ balls from $B''$ that contain the points of color $\ell$ in the optimal solution. Since $B$ has size $k = \sum_\ell k_\ell$, in the end $B''$ will have size at least $k_i \geq j^*$.

**Case 2:** $j < k$ **and** $t(j) = k$. Again in this case $B$ contains $k$ disjoint balls and one can obtain $B''$ with size at least $j^*$ similar to the above case.

**Case 3:** $j < k$ **and** $t(j) < k$. For this case, let us first prove the following simple claim.

**Claim A.3.** $r_{t(j)+1} < d_{j^*}^{i^*}/2$.

*Proof.* First note that by the definition of $t(j)$, we know that $r_{t(j)+1} < r_j/2$. Now if we have $r_j \leq d_{j^*}^{i^*}$ (this happens when we increased the value of $j$ in the last step), then we have that $r_{t(j)+1} < r_j/2 \leq d_{j^*}^{i^*}/2$ and the claim is proved. Otherwise, it means that $r_{j+1} < d_{j^*}^{i^*}/2$ (since we did not increase $j$ in Item 3). But then since $t(j) \geq j$, we have that $t(j) + 1 \geq j + 1$. Therefore, $r_{t(j)+1} < d_{j^*}^{i^*}/2$ and the claim is proved again. $\qquad\square$

Therefore, all the points in $P$ lie within a distance of at most $r_{t(j)+1} < d_{j^*}^{i^*}/2$ of one of the centers of the balls in $B$. For each point $q \in R$ let $N(q)$ be the closest center among the centers of $B$ to $q$. Note that for two points $q_1, q_2 \in R$, $N(q_1) \neq N(q_2)$ as otherwise their distance is less than $2r_{t(j)+1} < d_{j^*}^{i^*}$ contradicting the definition of the set $R$. Let $B''$ be the subset of the $j^*$ balls centered at $N(R)$. Note that for any other point in the optimal solution, i.e., $q \in \text{OPT} \setminus R$, it cannot lie inside any of the balls in the subset $B''$, as again it contradicts the definition of $R$. This means that the set $P \setminus B''$ contains enough points: i.e., $k_i - |R|$ points from color $i$, and $k_\ell$ points from color $\ell$ for each $\ell \neq i$. $\qquad\square$

Therefore, having Claim A.2 and using Lemma 3.2, we get that $|B'| \geq j^*/m$. Now, let $S_1, \ldots, S_m$ be the solution created at the iteration corresponding to $i$ and $j$ as specified above. That is, $S_i$ contains the centers of the balls in $B'$ returned by Approach 2, together with an arbitrary set of $k_i - |B'|$ points from $P_i \setminus B'$. Further, for each $\ell \neq i$, let $S_\ell$ contain an arbitrary set of $k_\ell$ points from $P_\ell \setminus B'$. This solution contains a set of $|B'| \geq j^*/m$ points such that no other point in $S$ lies within a distance of $r_{t(j)}/2 \geq d_{j^*}^{i^*}/8$. Now, to see the approximation, note that

$$
\begin{aligned}
\text{div}(\text{SOL}) &\geq \text{div}(S) \\
&\geq (j^*/m) \cdot d_{j^*}^{i^*}/c_1 \quad \text{by what we just proved} \\
&\geq \frac{1}{m \cdot c_1 \cdot \lceil \log(k_{i^*}+1) \rceil} \sum_{j' \leq k_{i^*}} d_{j'}^{i^*} \quad \text{by Claim 3.3} \\
&\geq \frac{1}{m^2 \cdot c_1 \cdot \lceil \log(k_{i^*}+1) \rceil} \sum_{i' \leq m} \sum_{j' \leq k_{i'}} d_{j'}^{i'} \quad \text{by choice of } i^* \\
&= \frac{1}{m^2 \cdot c_1 \cdot \lceil \log k \rceil} \cdot \text{div}(\text{OPT}) \quad \text{by definition of } d_{j'}^{i'}\text{'s}
\end{aligned}
$$

$\qquad\square$

## A.6 Proof of Theorem 4.1

*Proof.* We show the lemma by comparing the result of running Algorithm 3 on the union of the core-sets $S = \bigcup_i S_i$, and running it on the whole dataset $P = \bigcup_i P_i$. In particular, in Line 3 of Algorithm 3, running the GMM algorithm on $S_i$ would return the same set of points as running it on $P_i$, as $S_i$ itself contains $G_i$. Therefore, both runs of the algorithm end with the same set of balls $B$. We will now show the following claim.

**Claim A.4.** *Let $B$ be a set of at most $k$ disjoint balls of radius $r/2$ such that there exists a subset $B' \subseteq B$ of size at most $|B'| \leq k$ of them such that $P_\ell \setminus B'$ contains $k_\ell$ points. Further let $C$ be a set of $|B|$ balls with the same centers as $B$ but with radius $r/6$ instead, and let $C' \subseteq C$ be the subset of them corresponding to $B'$. Then $S_\ell \setminus C'$ contains $k_\ell$ points.*

*Proof.* Note that $S_\ell$ is the result of running the GMM algorithm $k$ times on $P_\ell$ each for $k+1$ iterations. Now, for $i \leq k$, let us use $T^i$ to denote the set of $k+1$ points returned by the $i$th run of GMM. Thus, $T^i$ is the result of running GMM on the set $P_\ell \setminus \bigcup_{j<i} T^j$. Let $r^i$ be the minimum pairwise distance between the points in $T^i$. Now if for all $i \leq k_\ell$, $T^i$ contains at least one point outside of the balls in $C'$ then clearly $T_\ell$ contains $k_\ell$ points outside $C'$ and the claim is proved.

Otherwise consider the first iteration $i$ such that $T^i \subset C'$. This means that since $|T^i| = k+1$ and $|C'| = |B'| \leq k$, there exists a pair of points in $T^i$ whose distance is at most $2 \cdot r/6 = r/3$. But then by Property 2 of GMM, this means that all the other points in $P_\ell \setminus \bigcup_{j \leq i} T^i$ are within a distance of $r/3$ from one of the balls in $C'$ and therefore, they are all within $B'$. This means that $P_\ell \setminus B'$ can only contain points from $\bigcup_{j<i} T^j \subseteq S_\ell$, and thus the lemma is proved. $\qquad\square$

This means that if we replace the set of balls $B$ in Algorithm 3 with $C$, both Approach 1 and Approach 2, will be able to find the set $C'$ (either exactly or approximately depending on which Approach we use). This means that corresponding to each solution SOL found by Algorithm 3 2, there will be a corresponding solution in the core-set of diversity at least $\text{div}(\text{SOL})/3$ (as the radius of the balls in

$C$ is $1/3$ of the radius of the balls in $B$). Therefore, as Theorem 3.5 shows that using approach 1, one gets a solution with approximation factor of $O(m \cdot \log k)$, then this means that Algorithm 4 provides a composable core-set of size $O(k^2)$ for each color, with an approximation factor of $O(m \cdot \log k)$. □

# B   Generalization of Section 2 to the case where $k_i \geq 1$

In Section 2 of the paper, for simplification of the proof, we assumed that for all $i$, we have $k_i > 1$. Here, we remove this assumption, and show that a simple modification of the algorithm works even if for some $i$'s we have $k_i = 1$. The modified algorithm is shown in Algorithm 5. The only modification made is that in Line 1, the GMM will be run for at least two iterations even if $k_i = 1$.

---

**Algorithm 5** Core-set Construction Algorithm for SUM-PAIRWISE

---

**Input** a point set $P_i$, together with parameters $k_i$ and $k$ (where $k = k_1 + \cdots + k_m$)
**Output** a subset $S_i \subseteq P_i$

1: $S_i = \{p_1, \ldots, p_{k_i}\} \leftarrow \text{GMM}(P_i, \max\{k_i, 2\})$
2: $T \leftarrow \emptyset$
3: **for** $p \in S_i$ **do**
4:     **for** $j = 1$ **to** $k_i$ **do**
5:         $T \leftarrow T \cup$ the closest point $p_j \in P_i \setminus T$ to $p$ s.t. $\arg\min_{q \in S_i} \text{dist}(p_j, q) = p$.
6:     **end for**
7: **end for**
8: $S_i \leftarrow S_i \cup T$
9: **return** $S_i$

---

**Theorem B.1.** *Algorithm 5 produces a core-set with size $O(k_i^2)$ and a constant factor approximation:* $\text{div}_{k_1,\ldots,k_m}(S) \geq \frac{1}{C} \cdot \text{div}_{k_1,\ldots,k_m}(P)$ *for a constant c.*

*Proof.* Again it is easy to see that the size of each $S_i$ is at most $k_i^2$, and we only need to show that the approximation factor is $O(1)$. We use the same notation as in the proof of Theorem 2.1. Note the only modification is that now for colors $i$ with $k_i = 1$, we have that $r_i$ is the distance between the two points in $S_i$ returned by $\text{GMM}(P_i, \max\{k_1, 2\})$ in Line 1 of the algorithm. Again, we divide the optimal value of the diversity into two parts $A$ and $B$ defined as before. Again we consider the two cases separately. The case of $A < B$ (i.e., Case 2 in the proof of Theorem 2.1), holds similar to before. So we only need to prove the case of $A \geq B$, which we show below.

In this case as before, we have that $\text{div}(\text{OPT}) \leq 20 \cdot k \sum_{i \in [m]} k_i r_i$.

Now let us divide the colors into two groups based on the value of their $k_i$, i.e., let $G = \{i \colon k_i > 1\}$ and thus $[m] \setminus G$ will contain colors $i$ with $k_i = 1$. WLOG, let us rename the colors so that the first $|G|$ colors of $[m]$ are the colors in $G$, and moreover, colors $|G| + 1$ to $m$ are sorted such that $r_{j_1} \leq r_{j_2}$ whenever $|G| + 1 \leq j_1 < j_2 \leq m$. Finally, we define $t = \sum_{j \leq |G|} k_j$ to be the total number of points in any solution that are from groups with at least one point in them.

**Constructing the solution.**   Now let us define a solution as follows.

- For $i \leq |G|$, define $\text{SOL}_i \subseteq S_i$ as before, to be the first set of $k_i$ points added to $S_i$ in Line 1 of the algorithm using GMM. Let $\text{SOL}^G = \bigcup_{i \in G} \text{SOL}_i$ be the union of all these solutions.

- To define $\text{SOL}_i$ for $|G| + 1 \leq i \leq m$, note that $k_i = 1$, and thus we need to decide which of the two points added in Line 1 of the algorithm to $S_i$ should be picked in $\text{SOL}_i$. Note that these two points are at distance exactly $r_i$.

  We go over the colors $|G| + 1$ to $m$ one by one and decide which of the two points to choose for $\text{SOL}_i$. When at color $i$, let $p_i, p_i' \in S_i$ be the two points returned in Line 1 of the algorithm using GMM which are at distance $r_i$ from each other. We let $\text{SOL}_i = \{p_i\}$ if $\sum_{x \in \bigcup_{j < i} \text{SOL}_j} \text{dist}(x, p_i) \geq \sum_{x \in \bigcup_{j < i} \text{SOL}_j} \text{dist}(x, p_i')$, and let $\text{SOL}_i = \{p_i'\}$ otherwise.

This way of constructing the solution gives us the property that

$$\sum_{x \in \bigcup_{j < i} \text{SOL}_j} \text{dist}(x, \text{SOL}_i) \geq \frac{r_i}{2} \cdot (t + i - 1 - |G|)$$

where we abused the notation $\text{SOL}_i$ to refer to the only point in $\text{SOL}_i$.

**Analysis.** To show that the constructed solution has large diversity, let us divide the participating distances into three parts as below.

$$\text{div}(\text{SOL}) = \sum_{x,y \in \text{SOL}^G} \text{dist}(x, y) + \sum_{x \in \text{SOL}^G, y \in \text{SOL} \setminus \text{SOL}^G} \text{dist}(x, y) + \sum_{x, y \in \text{SOL} \setminus \text{SOL}^G} \text{dist}(x, y)$$

Let us refer to the above three terms as $\text{div}_1$, $\text{div}_2$, and $\text{div}_3$. Now by the construction of $\text{SOL}_i$ for $i \leq |G|$, similar to Case 1 in the proof of Theorem 2.1, one can show that

$$\text{div}_1 + \text{div}_2 \geq \frac{k}{4} \sum_{i \leq |G|} k_i r_i$$

**Claim B.2.** $\text{div}_2 + \text{div}_3 \geq \frac{k}{8} \sum_{i=|G|+1}^{m} r_i$.

*Proof.* Note that we have

$$\text{div}_2 + \text{div}_3 = \sum_{x \in \text{SOL} \setminus \text{SOL}^G, y \in \text{SOL}} \text{dist}(x, y) \geq \sum_{i=|G|+1}^{m} \frac{r_i}{2} \cdot (t + i - 1 - |G|)$$

Now we consider two cases separately, if $t \geq k/2$ then clearly $(t + i - 1 - |G|) \geq k/2$ for $i \geq |G| + 1$, and we have

$$\text{div}_2 + \text{div}_3 \geq \sum_{i=|G|+1}^{m} \frac{r_i}{2} \cdot (t + i - 1 - |G|) \geq \sum_{i=|G|+1}^{m} \frac{r_i}{2} \cdot \frac{k}{2} = \frac{k}{4} \cdot \sum_{i=|G|+1}^{m} r_i$$

Otherwise, if $t < k/2$ then since we assumed that the $r_i$ for $|G| + 1 \leq i \leq m$ are sorted, and noting that the last term in the summation is $\frac{r_m}{2} \cdot (k - 1)$, we have

$$\text{div}_2 + \text{div}_3 = \sum_{i=|G|+1}^{m} \frac{r_i}{2} \cdot (t + i - 1 - |G|) \geq \sum_{i=m-k/2}^{m} \frac{r_i}{2} \cdot \frac{k}{2} = \frac{k}{4} \cdot \sum_{i=m-k/2}^{m} r_i \geq \frac{k}{8} \sum_{i=|G|+1}^{m} r_i$$

$\square$

Thus, noting that $k_i = 1$ for $i \geq |G| + 1$, and using the upper bound we stated on $\text{div}(\text{OPT})$, we get that

$$\text{div}(\text{SOL}) = \text{div}_1 + \text{div}_2 + \text{div}_3 \geq \frac{1}{2} \left( \frac{k}{4} \sum_{i \leq |G|} k_i r_i + \frac{k}{8} \sum_{i=|G|+1}^{m} k_i r_i \right) \geq \frac{1}{16} \sum_i k_i r_i \geq \frac{1}{320} \text{div}(\text{OPT})$$

$\square$

## C  Arbitrary Partitioning and Fully Composable Core-sets

In this section, we show what our results imply when the partitioning of the point sets into multiple parts is not necessarily based on the colors. This will provide a *composable core-set* as defined in [20] as opposed to the *color-abiding composable core-set* as defined in 1.4.

Formally, the goal is to present a summarization algorithm $\mathcal{A}$ that now receives a colored point set $P = \bigcup_{i \leq m} P_i$ and the goal is to summarize it into $T \subseteq P$ with the following composablility property. For any collection of colored point sets $P^{(1)}, \cdots, P^{(N)}$, we have that

$$\mathrm{div}_{k_1, \cdots, k_m}(\bigcup_{j \leq N} \mathcal{A}(P^{(j)})) \geq \frac{1}{\alpha} \mathrm{div}_{k_1, \cdots, k_m}(\bigcup_{j \leq N} P^{(j)})$$

**Corollary C.1.** *We have an algorithm that given a colored point set $P = \bigcup_{i \leq m} P_i$ computes a (not necessarily color-abiding) composable core-set of size $k^2$ for the Fair Diversity Maximization problem under the* SUM-PAIRWISE DIST *notion of diversity.*

*Proof.* The algorithm simply runs Algorithm 2 for each of $P_i$ and returns their union. The size of the core-set is clearly $\sum_{i \leq m} k_i^2 \leq k^2$. Moreover, by Theorem 2.1 it is easy to verify that this is in fact a composable core-set, because for any collection of colored point sets $P^{(1)}, \cdots, P^{(N)}$, one can treat them as having $N \cdot m$ different groups with $k_i^{(j)} \leq k_i$ (more precisely, one can set $k_i^{(j)}$ to be the number of points the optimal solution on $\bigcup_{j \leq N} P^{(j)}$ picks from $P^{(j)}$). Now note that although one does not know the value of $k_i^{(j)}$ at the time of running Algorithm 2 on $P_i^{(j)}$, however since $k_i^{(j)} \leq k_i$, it is easy to verify that running Algorithm 2 on $P_i^{(j)}$ with parameter $k_i$ as opposed to $k_i^{(j)}$ still works. $\square$

Similarly, one can get a composable core-set for the SUM-NN DIST notion of diversity.

**Corollary C.2.** *We have an algorithm that given a colored point set $P = \bigcup_{i \leq m} P_i$ computes a (not necessarily color-abiding) composable core-set of size $O(m \cdot k^2)$ for the Fair Diversity Maximization problem under the* SUM-NN DIST *notion of diversity.*

*Proof.* The algorithm simply runs Algorithm 4 for each of $P_i$ and returns their union. The size of the core-set is clearly $\sum_{i \leq m} k^2 \leq mk^2 = O(k^3)$. Moreover, by Theorem 4.1 it is again easy to verify that this is in fact a composable core-set, because for any collection of colored point sets $P^{(1)}, \cdots, P^{(N)}$, again one can treat them as having $N \cdot m$ different groups with $k_i^{(j)} \leq k_i$. Again, although one does not know the values of $k_i^{(j)}$ at the time of running Algorithm 4, however since $k_i^{(j)} \leq k_i$, it is easy to verify that the point set picked by running Algorithm 4 on $P_i^{(j)}$ with parameter $k_i$ is a super set of the core-set that would have been picked if the algorithm was run with parameter $k_i^{(j)}$ as opposed to $k_i$. $\square$

*Remark* C.3 (Applications). Now that we showed how to get a (not necessarily color-abiding) composable core-set, the result of [20] readily implies that we have a streaming algorithm and an MPC algorithm for the Fair Diversity Maximization under the SUM-PAIRWISE DIST and SUM-NN DIST notions of diversity.

# D  Data Preparation and Additional Experiments

In order to demonstrate the effectiveness of the proposed approximation algorithms, we run simulations on public and timed datasets. In this section, we give details on the used datasets, data preparation, pre-processing, and additional experiment results.

## D.1  Datasets, data preparation, and experiments setup

**Datasets and data preparation.**  We use *Reddit* public dataset [42] of text messages that are semantically embedded into a metric space (e.g., 748-dimensional metric space with BERT embeddings [12]). We use this collection[3] of Reddit messages as it provides the creation time stamp in the schema, which we need to mark the relative recency of the messages, presented by its color

---

[3]`https://github.com/henghuiz/MaskedHierarchicalTransformer`

within a given month[4]. We want to remark other commonly used Reddit datasets [37, 22][5] are not considered in this work as they do not have the creation time in their schema. Since we have the messages metadata included in the creation timestamp $t_i$ of a message $i$, we can assign a color $color_i$ of message $i$ as

$$color_i = \lfloor m \cdot \frac{t_i - t_{\min}}{t_{\max} - t_{\min}} \rfloor$$

where $t_{\min}$ and $t_{\max}$ are the minimum and maximum creation timestamp within the considered full input data and $m$ is the desired total number of colors. In this way, we have $m$ equidistant time intervals such that messages (relatively) close to one another by creation time belong to the same color and the most recent messages are in the $m$-th color. In this way, we can also run Fair Diversity Maximization (FDM) for a desired distribution of number of messages within a color, i.e., $k_i$, for $\forall i = 1, 2, \ldots, m$, either by *fairness* (all $k_i$'s equal) or by recency ($k_i$ increases with $i$). In order to transform the message into a multi-dimensional space, we use BERT-model semantic embeddings [12] in our experiments (presented in the paper). Sentences and paragraphs embedding is an active area of NLP research [34, 36]. Although, the quality of sentences and paragraphs semantic embedding is out-of-scope for this work and we assume the messages are already given in the metric space, we have demonstrated (in Section D.2.4) the trend of the results is preserved by using other state-of-the-art (such as MPNETv2 [35], T5 [29], RoBERTa [23] or MiniLM [38]) and traditional (like GloVe [32]) pre-trained models for semantic embeddings. In the paper, the input "pool" of messages for the DM and FDM algorithms in the experiments is set to one month as we want to present to a user messages that are selected from a timely and fresh period and the DM and FDM approximation algorithms make a decision for the most relevant, but also diverse and fairly distributed (per color) messages. We present results from January 2018 Reddit dataset that consists of 21,474 samples, divided into four colors[6] and our experiments show there is a consistency in the results using input data from other periods.

We also utilize the *MovieLens* movie reviews dataset [18] from 2001, where the movie titles are semantically embedded into a metric space. In addition to the color determined by the creation time of the review (determined in the same way as explained for the *Reddit* dataset), for *MovieLens* dataset, we also assign a color based on the movie genre. There are 18 movie genres: {'Western', 'Documentary', 'Mystery', 'Film-Noir', 'Thriller', 'Crime', 'Adventure', 'Musical', 'Sci-Fi', 'War', 'Action', 'Fantasy', 'Comedy', "Children's", 'Animation', 'Horror', 'Romance', 'Drama'}.

**Description of the experiments.** The main experiments we run in this work are the followings.

- First we show that if we run DM on the data, the results are not balanced as we want, see Section 5 (main part, Figure 1) and Section D.2.1 (Figures 2 and 3), and thus we need to resort to FDM. Further, we show that using FDM the diversity does not decrease by much. This is shown in Section 5 (main part, Table 2) and Section D.2.2 (Tables 4 and 5).

- Second, we show the effectiveness of the core-sets. We run our core-set construction algorithms (developed in Section 2 and Section 4) on each color independently, to get a smaller size dataset. We then run the FDM optimization once on the union of the core-sets and once on the whole data. We measure the diversity loss and runtime improvement achieved by the use of core-sets. The results are given in Section 5 (main part, Table 3) and Section D.2.3 (Table 6).

- The difference of the results (in the SUM-PAIRWISE notion of diversity) if we use alternative message embeddings are given in Section D.2.4 (Table 7).

- Last, the practical comparison of our algorithm with the state-of-the-art algorithm of Ceccarello *et al.* [8] for core-set construction respect with to SUM-PAIRWISE diversity is given in Section D.2.5 (Table 8).

---

[4]The messages in this dataset are given with a creation time (time stamp) and message IDs, but with no content / message text and the message content is extracted by using the Reddit API: `https://github.com/reddit-archive/reddit/wiki/OAuth2`. We request the messages for several months (using a similar script as the one used by the dataset creators) as we want to provide fresh messages from the recent month (our search pool) in our output.

[5]see also: `https://www.tensorflow.org/datasets/catalog/reddit` and `https://huggingface.co/datasets/reddit_tifu`

[6]Based on the quarter (roughly a week) to which a message belongs within this month.

## D.2 Additional experiment results

### D.2.1 Fairness of the DM algorithms outcome and justification of FDM need

While the DM algorithms produce diverse results, they often do not have the balance in the fairness we prefer to have. When we make the color division based on time periods ($m = 4$, quarters within a month), DM algorithms produce imbalanced outcome ($k = 20$) for colors presence, see Figure 1 (main part) and Figure 2. The conclusions are two-fold: (i) the most present messages are not the most recent ones and (ii) the results are also not equally balanced per quarter. In the case of *MovieLens* where the color is the movie genre ($m = 18$ genres), the most present movies genres in the most optimal $k = 50$ items are *Drama* and *Comedy* across all diversity distances and some genres are not present as shown in Figure 3. This highlights the need for the colored version of the algorithms (FDM) to achieve the desired fairness.

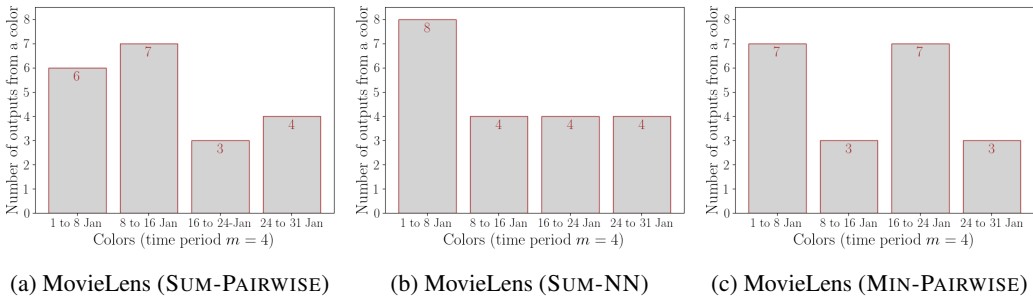

(a) MovieLens (SUM-PAIRWISE)     (b) MovieLens (SUM-NN)     (c) MovieLens (MIN-PAIRWISE)

Figure 2: DM algorithm outcomes for all colors ($m = 4$) with equidistant time period as fairness colors in the MovieLens dataset ($k = 20$ items).

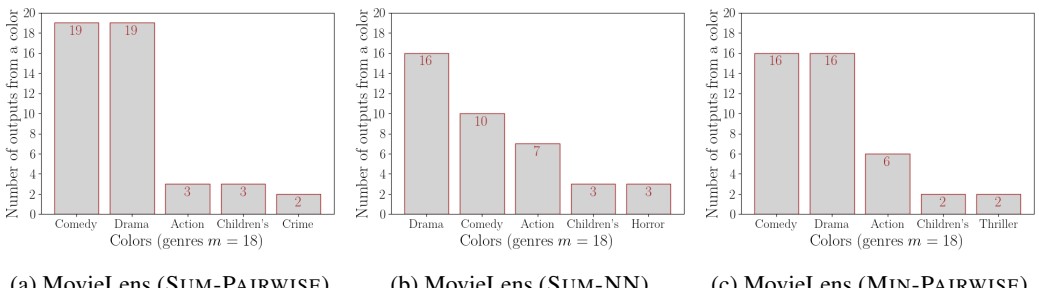

(a) MovieLens (SUM-PAIRWISE)     (b) MovieLens (SUM-NN)     (c) MovieLens (MIN-PAIRWISE)

Figure 3: DM algorithm outcomes for top 5 colors with genres as fairness colors ($m = 18$) in the MovieLens dataset ($k = 50$ items).

### D.2.2 DM vs. FDM results

**Reddit dataset.** When we run FDM, we gain in fairness (or balancedness), but we lose in diversity over the DM expressed in different metrics. How much the diversity becomes worse when we apply the FDM, for the Reddit dataset, is shown in Table 2. Experiments show that we lose around 1% for SUM-PAIRWISE DIST; from few percent up to no more than 20% for SUM-NN DIST and around 50% for MIN-PAIRWISE DIST (due to its fragility) for various distributions per color, while achieving the desired per colors distribution, both in balancedness (uniform $k_i = const$) and recency ($k_i$ increasing linearly; $k_m$ most recent items).

**MovieLens dataset.** The results for the *MovieLens* dataset follow the trend of the results of Reddit dataset, when we compare the diversity loss (and fairness gains). This is given in Table 4, comparable to Table 2 in the main part of the paper. In the cases with time as a color, experiments show that on diversity, we lose less than 1% for SUM-PAIRWISE DIST; few percents for SUM-NN DIST and around 50-70% for MIN-PAIRWISE DIST, while achieving the desired per color distribution. For this dataset, we also run experiments where the number of colors is larger (genres; $m = 18$) and we also

see a similar trend of diversity loss of around 2-3% for for SUM-PAIRWISE DIST; few percents for SUM-NN DIST and around 50-55% for MIN-PAIRWISE DIST as shown in Table 5.

Table 4: The *loss* of diversity (% Div. loss) between DM vs. FDM on the full data, expressed as a relative change, of the concerned SUM-PAIRWISE, SUM-NN and MIN-PAIRWISE distances for uniform (upper part) or increasing (lower part) color values $k_i$ for *MovieLens* dataset with time period within January 2001 as colors ($m = 4$).

| DM vs. FDM | | SUM-PAIRWISE | SUM-NN | MIN-PAIRWISE |
|---|---|---|---|---|
| colors $k_i$ | $\sum k_i$ | % Div. loss | % Div. loss | % Div. loss |
| $[2, 4, 6, 8]$ | 20 | 0.14% | 1.46% | 58.93% |
| $[3, 6, 9, 12]$ | 30 | 0.02% | 3.33% | 68.94% |
| $[4, 8, 12, 16]$ | 40 | 0.03% | 1.62% | 68.32% |
| $[5, 10, 15, 20]$ | 50 | 0.02% | 1.70% | 67.64% |
| $[6, 12, 18, 24]$ | 60 | 0.02% | 1.15% | 67.36% |

Table 5: The *loss* of diversity (% Div. loss) between DM vs. FDM on the full data, expressed as a relative change, of the concerned SUM-PAIRWISE, SUM-NN and MIN-PAIRWISE distances for uniform color values $k_i$ for *MovieLens* dataset with genres as colors ($m = 18$).

| DM vs. FDM | | SUM-PAIRWISE | SUM-NN | MIN-PAIRWISE |
|---|---|---|---|---|
| colors $k_i$ | $\sum k_i$ | % Div. loss | % Div. loss | % Div. loss |
| $[2, 2, \ldots, 2]$ | 36 | 2.84% | 9.51% | 52.63% |
| $[3, 3, \ldots, 3]$ | 54 | 2.88% | 5.78% | 54.04% |
| $[4, 4, \ldots, 4]$ | 72 | 2.87% | 11.38% | 53.09% |
| $[5, 5, \ldots, 5]$ | 90 | 2.93% | 12.52% | 52.25% |
| $\underbrace{\phantom{xxxxx}}_{18}$ | | | | |

### D.2.3 Effectiveness of the core-sets results

**Reddit dataset.** We also run experiments to show the power of data summarization in terms of core-sets. The results in terms of diversity loss are on-par or with marginal difference if we apply the FDM to the union of core-sets, compared to FDM applied to the full data. See Table 3 for the results on both SUM-PAIRWISE and SUM-NN distances. (In the case of SUM-NN DIST, there are cases where one is marginally better than the other[7].) Moreover, running FDM to the union of core-sets is orders of magnitude faster compared to the same algorithm applied to the full data.

**MovieLens dataset.** As shown in Table 6, the comparison of the diversity outcome of the FDM applied on the full data *versus* the FDM applied on the data summary presented by the core-sets show similar trend as in the Reddit dataset (Table 3): few percent in diversity is lost for SUM-PAIRWISE notion, while for the SUM-NN the results are mixed either in favor of the full data outcome or the core-set in terms of diversity. In the case of MIN-PAIRWISE the loss of diversity is either 0% in a case the two points connected with the minimum distance are in the union of the core-sets, or it can be a higher value in the opposite case due to the fragility of the MIN-PAIRWISE DIST.

---

[7]For example, FDM on the core-sets is occasionally better in diversity compared to the full data. One should note that the higher approximation factor contributes to this.

Table 6: The *loss* of diversity (% Div. loss) expressed as a relative change of diversity distances, and the running time *gains* ($\times$ times faster) of the FDM when applied to the union of core-sets compared to FDM applied to the full data for SUM-PAIRWISE, SUM-NN and MIN-PAIRWISE distances for uniform or increasing color values $k_i$ for *MovieLens* dataset for January 2001 and $m = 4$.

| FDM full data vs. core-sets | | SUM-PAIRWISE | | SUM-NN | | MIN-PAIRWISE | |
|---|---|---|---|---|---|---|---|
| colors $k_i$ | $\sum k_i$ | % Div. loss | Time gain ($\times$) | % Div. loss | Time gain ($\times$) | % Div. loss | Time gain ($\times$) |
| $[2, 2, 2, 2]$ | 8 | 1.16% | 1 515.18 | 2.81% | 1 225.64 | 0.00% | 1 734.86 |
| $[3, 3, 3, 3]$ | 12 | 1.38% | 967.96 | $-0.16\%$ | 604.15 | 26.51% | 1 130.37 |
| $[4, 4, 4, 4]$ | 16 | 0.79% | 617.67 | $-1.62\%$ | 325.73 | 46.17 | 874.17 |
| $[5, 5, 5, 5]$ | 20 | 0.61% | 455.46 | 0.00% | 207.71 | 26.75 | 686.42 |
| $[6, 6, 6, 6]$ | 24 | 0.62% | 277.63 | $-0.23\%$ | 128.25 | 26.75 | 582.37 |
| $[2, 4, 6, 8]$ | 20 | 0.83% | 412.02 | $-2.94\%$ | 205.90 | 9.25% | 1 515.18 |
| $[3, 6, 9, 12]$ | 30 | 0.77% | 143.14 | 0.04% | 83.48 | 0.68% | 967.96 |
| $[4, 8, 12, 16]$ | 40 | 0.90% | 121.84 | 5.30% | 41.73 | 11.79% | 617.67 |
| $[5, 10, 15, 20]$ | 50 | 0.70% | 30.68 | 0.05% | 25.92 | 0.00% | 455.46 |
| $[6, 12, 18, 24]$ | 60 | 0.75% | 18.40 | $-8.85\%$ | 13.07 | 43.38% | 277.63 |

### D.2.4 Experiment results with alternative semantic embeddings

In this part, we show that using different embeddings for the messages produce similar trend in the obtained results. Table 7 present the comparison results in terms of diversity loss in the notion of SUM-PAIRWISE DIST if we apply a FDM algorithm in comparison with the standard DM algorithm for various semantic message embeddings. The results show that we have a negligible loss of diversity, while we have the desired fairness (in terms of the requested $k_i$'s); irrelevant of the employed pre-trained model to embed the messages in a multi-dimensional space by using 6 state-of-the-art pre-trained models for embedding a message or a paragraph: BERT [12]; T5 [29]; MPNETv2 [35]; MiniLM [38]; RoBERTa [23] and GloVe [32]. We use one of the highest performance and popular pre-trained models (without further training), however the quality of semantic sentence & paragraph embeddings using Large Language Models (LLM) is out-of-scope for this work.

### D.2.5 Comparison of core-set construction algorithms

In this section for the SUM-NN notion of diversity, we first compare the size of the obtained core-sets of the algorithm proposed in this paper and the one from [8], in practice, for Reddit data. The results are given in Table 8. The results show that core-set size of the algorithm from [8] can indeed be very high providing very little summarization, while the core-set of our proposal is indeed much smaller (columns 3 and 4). As a result, our algorithm is few orders of magnitude faster than the core-set construction algorithm from [8] (column 5). This is true while the diversity loss of our algorithm is negligible (less than 1% to few percent).

Table 7: The *loss* of diversity (% Div. loss) between DM vs. FDM on the full data, expressed as a relative change, of the concerned SUM-PAIRWISE DIST for uniform or increasing color values $k_i$ for six popular semantic embeddings. The results in the main part of the paper and in the other figures in this appendix are based on BERT [12] message embeddings.

| DM vs. FDM | | BERT [12] | T5 [29] | MPNETv2 [35] |
|---|---|---|---|---|
| colors $k_i$ | $\sum k_i$ | % Div. loss | % Div. loss | % Div. loss |
| $[2, 2, 2, 2]$ | 8 | 1.22% | 1.15% | 0.32% |
| $[3, 3, 3, 3]$ | 12 | 0.98% | 0.60% | 0.32% |
| $[4, 4, 4, 4]$ | 16 | 0.50% | 0.30% | 0.20% |
| $[5, 5, 5, 5]$ | 20 | 0.47% | 0.29% | 0.11% |
| $[6, 6, 6, 6]$ | 24 | 0.19% | 0.20% | 0.05% |
| $[2, 4, 6, 8]$ | 20 | 0.42% | 0.26% | 0.08% |
| $[3, 6, 9, 12]$ | 30 | 0.29% | 0.23% | 0.14% |
| $[4, 8, 12, 16]$ | 40 | 0.25% | 0.08% | 0.06% |
| $[5, 10, 15, 20]$ | 50 | 0.16% | 0.10% | 0.02% |
| $[6, 12, 18, 24]$ | 60 | 0.12% | 0.10% | 0.03% |
| | | MiniLM [38] | RoBERTa [23] | GloVe [32] |
| | | % Div. loss | % Div. loss | % Div. loss |
| $[2, 2, 2, 2]$ | 8 | 1.14% | 1.27% | 1.11% |
| $[3, 3, 3, 3]$ | 12 | 0.71% | 0.17% | 0.47% |
| $[4, 4, 4, 4]$ | 16 | 0.38% | 0.06% | 0.13% |
| $[5, 5, 5, 5]$ | 20 | 0.32% | 0.14% | 0.03% |
| $[6, 6, 6, 6]$ | 24 | 0.21% | 0.05% | 0.10% |
| $[2, 4, 6, 8]$ | 20 | 0.38% | 0.31% | 0.43% |
| $[3, 6, 9, 12]$ | 30 | 0.35% | 0.18% | 0.38% |
| $[4, 8, 12, 16]$ | 40 | 0.27% | 0.19% | 0.31% |
| $[5, 10, 15, 20]$ | 50 | 0.32% | 0.14% | 0.32% |
| $[6, 12, 18, 24]$ | 60 | 0.33% | 0.15% | 0.13% |

Table 8: The size of the union of core-sets produced by the algorithm of [8] and the size of the union of core-sets produced by our algorithm. The *loss* of diversity (% Div. loss) expressed as a relative change of diversity distances, and the running time *gains* ($\times$ times faster) of the FDM when applied to the union of core-sets produced by our algorithm vs. the one from [8] for SUM-PAIRWISE distance for uniform or increasing color values $k_i$ for the Reddit dataset.

| Corsets size comparison | | $\cup_i$ core-sets size | | core-sets compute | FDM ($\cup_i$ core-sets) | |
|---|---|---|---|---|---|---|
| colors $k_i$ | $\sum k_i$ | [8] alg. | our alg. | Time gain ($\times$) | % Div. loss | Time gain ($\times$) |
| $[2, 2, 2, 2]$ | 8 | 12791 | 40 | 17.19 | 3.89% | 1 055.62 |
| $[3, 3, 3, 3]$ | 12 | 13486 | 56 | 9.20 | 0.62% | 529.96 |
| $[4, 4, 4, 4]$ | 16 | 13857 | 72 | 5.28 | 0.26% | 458.05 |
| $[5, 5, 5, 5]$ | 20 | 13993 | 154 | 3.65 | 1.69% | 246.58 |
| $[6, 6, 6, 6]$ | 24 | 14305 | 164 | 2.78 | 0.27% | 129.73 |

