# OpenReview forum: "Core-sets for Fair and Diverse Data Summarization"
_NeurIPS.cc/2023/Conference — NeurIPS 2023 poster_

### Official Review · Reviewer_aN6A · 2023-07-05

**Soundness:** 2 fair
**Presentation:** 2 fair
**Contribution:** 3 good
**Rating:** 6
**Confidence:** 3

**Summary:**

The study presents improved core-set construction algorithms for maximizing diversity under partition constraints.
Two diversity measures were considered: sum-of-pairwise distances and sum-of-nearest-neighbor distances.
The algorithms are validated through real-world application to time-sensitive message data, resulting in a 100x speedup and minor diversity loss while improving space usage in streaming settings.

**Strengths:**

Algorithms 2, 3, and 4, despite their relative simplicity, provide significant insights, primarily stemming from the proofs of approximations related to their definitions.
These algorithms' straightforward yet effective design and the resultant analytical insights shed new light on the problem. These aspects, therefore, make these algorithms especially appealing to a broad research community. Conference attendees will find these insights valuable, sparking potential discussions around their contributions and their implications in the field.

**Weaknesses:**

While the paper offers some contributions, its structure requires revision for improved clarity, particularly moving the detailed 'Overview of the Algorithms' from the first section to the respective sessions dealing with the algorithms and balancing the disproportionate length of the first section within the overall context.

The renaming of 'Diversity Maximization under Partition Constraints' to 'Fair Diversity Maximization' is somewhat perplexing. The 'fairness constraints' applied in this study do not differ significantly from the 'partition constraints' defined in the existing literature. Moreover, in other works (e.g., [22]), the term' fairness' is applied based on specific definitions like group fairness or statistical parity. The decision to redefine 'partition constraints' as' fairness constraints' without a clear distinction is somewhat confusing, making it more difficult to discern the precise contributions of this paper.
Still, regarding the use of "fairness constraints" throughout the paper, the paper presents two main experiments. The first involves grouping a dataset of messages by creation time to compute a diverse summary that favors recently over older messages. The second uses the method to control each genre's contribution to a movie recommendation system. However, neither of these experiments demonstrates the necessity of applying fairness constraints (as defined in the literature) to solve the proposed problem.

The GMM Algorithm, a fundamental building block of the method, must be correctly defined. Since it initially selects some arbitrary point as the first element of S, [11]'s definition is accurate. In fact, because S begins as an empty set, the execution of line 3 in the authors' definition is inconsistent.

The text contains some English usage errors that affect clarity. Specifically, the phrase "based on greedy" found in line 107 could benefit from a more explicit definition. The part containing "along the way" in line 101 is vague and could be replaced with a more descriptive phrase. Furthermore, key terminologies, such as the "aspect ratio" of the dataset mentioned in lines 9, 86, and 94, are misused or not sufficiently defined.

Some details of the experiments in the main text, such as the choice of 21,474 samples mentioned in line 294, appear arbitrary, and the rationale for their selection is buried in the appendix.
Finally, the paper would greatly benefit from maintaining consistent naming and notation across all sections, enhancing its comprehensibility and overall cohesiveness.
For example, the "div" definition in line 231 should use the same notation as in line 238; many references to core-set vs. coreset.

**Questions:**

Section 1.3 defines the problem as selecting subsets S_i of k_i points from P_i to maximize div(S). However, the subsets S_i don't strictly contain k_i points in your final results. Instead, they include the core-set and the closest k_i points to each seed, effectively increasing the core-set size to k_i^2, as observed in the 'Core-set for SUM-PAIRWISE DIST' case. Could you please clarify how these results align with your original problem definition? Furthermore, could you please provide additional detail in your problem definition to help readers understand how the algorithm's results can serve as a potential solution to the problem?

**Limitations:**

The authors adequately addressed the limitations. There are no potential negative societal impacts from their work.

---

> ### Author Rebuttal · Authors · 2023-08-05
>
> Thank you for your positive feedback mentioning that our paper provides significant insights, effective algorithms, sound theoretical results supported by experiments, and, indeed, it can be of broader interest of the NeurIPS community and the conference attendees. Also thank you for all the feedback to improve the presentation of the paper. In what follows we address your comments in more detail.
>
>
> > Regarding the organization of Section 1 and having “overview of the algorithms” subsection
>
> Thank you for your suggestion on distributing the “overview of the algorithms” over to the corresponding sections. We believe this is more of a style of writing (e.g., this is very common in the theory community) . We are happy to follow your suggestion and will move the content of the section “overview of the algorithms” to the corresponding sections to balance the length of different sections.
>
> > Regarding your comment on the notions of “fairness constraint” vs. “partition constraint”
>
> Both notions are common in the literature. In fact [22] and [23] use “Diverse Data Selection under Fairness Constraints” to refer to the exact same problem of FDM under the minimum pairwise distance notion of diversity. We use the same grouping constraint as they do.
>
> Although we use “fairness constraint” similar to [22] and [23], we are explicitly referring to the term “partition constraint” as early as the second paragraph in the introduction. Again we are happy to rename all instances of “fairness constraint” to “partition constraint” per your suggestion.
>
> > Regarding your comment on the initialization of the greedy (GMM) Algorithm 1
>
> Thanks, we fixed this. Please check the PDF one-pager attached.
>
> > Regarding your minor comments on putting more details on dataset size in the main body, typos, and language issues:
>
> Thank you, we will also add more explanation around the datasets from the appendix to the main body (done this way due to space constraints). We will address the language issues that you mentioned and fix the typos with respect to “div”, and be consistent using “core-set” term. We will correct the the terms “based on greedy” and “along the way”. We will also define the aspect ratio clearly in the final version.
>
> > Regarding your comment / question on the definition of core-sets.
>
> **Definition 1.1** is the definition of diversity maximization under fairness/partition constraint. Here, the goal is to compute an **approximate solution** $S_i\subset P_i$ such that
>
> 1) $\|S_i\|$ is exactly $k_i$.
> 2) The overall diversity of $\bigcup_i S_i$ which consists of exactly $k=\sum_i k_i$ points is maximized.
>
> **Definition 1.4** on the other hand which is the focus of this paper defines **composable core-sets** for FDM. The goal here is to summarize each $P_i$ independently and get a core-set $T_i \subset P_i$ such that
>
> 1) Size of the core-set i.e., $\|T_i \|$ is small.
> 2) The union of the core-sets $T=\bigcup_i T_i$ contains a good solution.
>
> Once we have a good core-set construction algorithm, one can run any exact or approximate algorithm on the union of the core-sets/summaries as post-processing and compute the final solution on (e.g. one can use the algorithm of [24] referring to Table 1) on $T$.
>
> We will expand these definitions and put more details on them in the revision.
>
>
> >Although the findings from the study may be of interest to conference attendees, the issues identified indicate that the manuscript is not ready for publication in its current form. Hence, my recommendation is a borderline rejection of this manuscript. This manuscript does not meet the necessary standards for publication and requires substantial revisions
>
> Dear reviewer aN6A, once again we sincerely thank you for pointing us to places for improving the manuscript. While we agree with all your comments, we believe all of them are easily addressable and not a strong basis for “borderline reject”. We will further check the paper’s readability and structure.
>
> Could you please consider increasing your score given that we addressed all the above concerns and will apply the corresponding changes to the final draft?

---

> > ### Author Response · Authors · 2023-08-17
> > **Follow up on the rebuttal**
> >
> > Dear reviewer aN6A,
> >
> > We wanted to ask if we have addressed your concerns satisfactorily or if you have further concerns. Could you kindly let us know so that we have a chance to respond before the deadline? Thank you very much.

---

> > ### Comment · Reviewer_aN6A · 2023-08-17
> > **After authors' response**
> >
> > Firstly, I'd like to thank the authors for their detailed responses to my review. I decided to increase my score based on the clarifications provided in your comments. Additionally, after reviewing the feedback from other reviewers and your corresponding responses, I have gained a more comprehensive understanding that positively influences my assessment.

---

> > > ### Author Response · Authors · 2023-08-17
> > > **Follow up with Reviewer aN6A**
> > >
> > > Thank you for following up on our rebuttal and your response.

---

### Official Review · Reviewer_9Vmw · 2023-07-07

**Soundness:** 3 good
**Presentation:** 1 poor
**Contribution:** 3 good
**Rating:** 6
**Confidence:** 1

**Summary:**

A set of metric data points is comprised of m disjoint groups. The manuscript seeks for each group a subset of some given size such that the union of subsets the minimal pairwise distance (or sum of pairwise / nearest neighbour distances).  For constant factor approximation of minimising the sum of pairwise distances Section 2 of the manuscript reports a union of core-sets per group which for each group is quadratic in size. This result is achieved by applying Gonzalez's algorithm for each group and then simple post-processing steps over the obtained cluster centroids. Further results are achieved for sum of nearest neighbour distances as a diversity measure.

**Strengths:**

S1) Challenging and natural problem: fairness constraint akin to stratified sampling to achieve good group representation and diversity constraint to achieve good overall representation

S2) Non-trivial analysis: size of core-sets for min-pair-sum independent of the data size and distribution unlike in prior work.

S3) Practical: analysed methods are simple and sufficiently efficient for practical purposes

S4) Presentation is for the most part clear and prior work seems to be thoroughly discussed and compared against.

**Weaknesses:**

W1) Presentation could be clearer in some key points

Fairness constraint is not fully clear. The empirical results seem to indicate that the sizes per group dictate the sizes of the core-sets, but other parts of the paper indicate that the size of the core-sets could be larger than the dictated group sizes. Are the core-set sizes meant to be the intermediate sizes before the post-processing step?

**Questions:**

Q1) How exactly does the fairness constraint relate to the core-set / "solution" sizes and how can the core-sets be larger than the group sizes specified in the fairness constraints? (see W1)

**Limitations:**

Relevant limitations seem to be discussed.

---

> ### Author Rebuttal · Authors · 2023-08-05
>
> We thank you for your review and positive feedback.
>
> Precisely as you say, the goal in our paper is to provide an intermediate summary (i.e., a core-set) such that the union of the core-sets contains a good solution relative to the whole data. Later on, as a post-processing, one can use any exact or approximation algorithm on the union of the computed summaries to get a final solution. This is what we do in our experiments too.
>
> Please refer to Definition 1.4 and Definition 1.1 for exact definition of our setting. In particular we provide an algorithm which given a point set $P_i$, it computes a core-set $T_i$. The size of the core-set $T_i$ should be **low but not necessarily $k_i$**. We want our summaries to have the property that
>
> $div_{k_1,...,k_m}(T) \geq div_{k_1,...,k_m}(P)$
>
> This means that the union of the core-sets, i.e., $T=\bigcup_i T_i$ should contain at least $k_i$ points from color $i$ such that their overall diversity is relatively large in comparison to the optimal set of points one could have picked out of the whole data, i.e., $P$.
>
> We will add more explanation to clarify this distinction right after formal Definitions 1.1 and 1.4 in the final version.
>
> Dear reviewer 9Vmw, thanks again for the comments that improve the manuscript. Considering you stated the presentation is clear in most of the parts (S4) and with the newer clarifications that we will add around key concepts, would it be possible to reconsider your score for a higher (and the presentation score)?

---

> > ### Comment · Reviewer_9Vmw · 2023-08-17
> >
> > Thanks, it follows my rough understanding based on the reply and a quick revisit of the manuscript. The abstract states:
> >
> > - "Given a set of points $P$ in a metric space partitioned into $m$ groups, and given $k_1, . . . , k_m$, the goal of this problem is to pick $k_i$ points from each group $i$ such that the overall diversity of the $k = \sum_i k_i$ picked points is maximized."
> >
> > A similar problem is described in Definition 1.1. The goal in this manuscript is to build a core-set $T$ that may no longer contain the exact solution of the problem in Definition 1.1., but is guaranteed to contain an $\alpha$-approximate solution (cf. Definition 1.4) in terms of the diversity measure. The core-set $T$ in this work is the union of group-wise core-sets $T_i$ for each group $i$, i.e., $T = \bigcup_i T_i$.
> >
> > As an $\alpha$-approximate solution still needs to have $k_i$ points from each group, this follows that $T$ must have at least $k_i$ elements per color, but does not preclude having additional elements as the core-set is an intermediate result that can contain more than $k$ elements (e.g., achieved core-set sizes reported in Table 8 of the appendix). As in other core-set tasks the goal is to (approximately) solve some optimisation problem more efficiently by first narrowing it down to the most relevant data points. Table 3 reports the time gains and approximation quality (diversity loss/gain) achieved by running FDM on the core-set $T$ rather than the full set of points $P$, where the time gain implies that the core-set $T$ is much smaller than $P$ and the approximation quality implies that the core-sets contains the elements needed to build an approximate solution.

---

> > > ### Author Response · Authors · 2023-08-17
> > > **Follow up with Reviewer 9Vmw**
> > >
> > > Dear reviewer 9Vmw,
> > >
> > > Precisely, your understanding in your last message is fully correct. We very much appreciate your follow up with our rebuttal. We are happy to add these clarifications around the key concepts to make the presentation more clear in the final version.

---

### Official Review · Reviewer_S4pc · 2023-07-12

**Soundness:** 3 good
**Presentation:** 3 good
**Contribution:** 3 good
**Rating:** 6
**Confidence:** 4

**Summary:**

This paper gives coresets for two different fair diversity measures (FDM). Fairness is defined as a partition matroid where the number of items from each part is fixed. The authors give a coreset for FDM under the sum-pairwise distance. They also give a coreset for sum-NN-distance. The proposed algorithms are compared against baselines for a real dataset. The main technique appears to be a based on the greedy k-center clustering (which the authors term as GMM).


**Strengths:**

 Coresets for these measures have not been studied, hence these are new results. Based on the experiments, the results are fairly practical. We get both a theoretical guarantee and a practical boost in performance.

In the experiments, the authors compare with existing baselines to show that their coresets are much smaller while giving comparable performance.  While the technical contribution relies mostly on the k-center algorithm, I like the fact that the algorithms are simple.

**Weaknesses:**

 The theoretical guarantee for the NN-distance diversity notion seems fairly weak, O(m log k). We already have an O(1) approximation guarantee algorithm anyway, so it is unclear why this is useful.

**Questions:**

1. Why is the O(m log k) coreset guarantee useful?

**Limitations:**

The theoretical guarantees are not very impressive.

---

> ### Author Rebuttal · Authors · 2023-08-05
>
> We thank you for reviewing our paper and for the kind words on the paper merit.
>
> We want to clarify that the existing O(1) approximation is in the “standard offline setting” of the problem and it **does not** give a “core-set construction algorithm” (Please refer to Table 1). This is because those O(1) approximation algorithms assume to have access to the whole data at hand which is not the setting we consider. We will ensure to make this more clear in the final version of the paper. In our setting, each pointset corresponding to a group should be processed and summarized independently.
>
> More precisely, the goal in the core-set setting is to find a “small summary” $T_i$ for each point set $P_i$ such that the union of the core-sets $T=\bigcup_i T_i$ contains a good solution compared to the whole data $P=\bigcup_i P_i$. Please refer to Definition 1.4 and Definition 1.1.
> While the goal of this paper is to present a good summarization algorithm for computing $T_i$ from $P_i$, the goal of existing O(1) approximation is for computing an approximate solution (on either $P$ or $T$).
>
> Moreover, in a practical scenario we have (e.g., news or social network feed), $k$ is in order of 10s (e.g, 20 or 50) and $m$ is often a single digit or in order of 10s in the worst case, thus the O($m \log k$) can still be useful as also shown by our experiments.

---

### Official Review · Reviewer_CoyK · 2023-07-31

**Soundness:** 4 excellent
**Presentation:** 3 good
**Contribution:** 3 good
**Rating:** 7
**Confidence:** 4

**Summary:**

This paper investigates data summarization problems.  Given a set of points in metric space, the goal is to select k points that are diverse. Diversity could be measured as the sum of squared distances, the sum of nearest neighbor distances, etc.

Constant approximation algorithms for these problem are known.   This paper investigates core-sets for the problem.  A shortcoming of prior work is that say given two data sets that have been summarized, how can one combine their summarizations? Simply taking two solutions and combining them can give poor performance.   This paper offers coresets that are composable; that is, can be combined.

The core sets are of size poly(k), independent of n. The approximation ratio is constant for the sum of pairwise distances and is O(m log k) when there are m classes for the NN metric.  The ideas in the paper are simple natural approaches to the problem.

A clear application of the core-sets is to get faster algorithms for big data sets.  Indeed, this approach naturally leads to much better run times practically.

Overall, the paper is reasonable. On one hand, they theoretical ideas are simple. On the other, it seems to me that these results should be known and they clearly make the problem scalable to larger data sets.

Comment:

The paper would be even stronger by pointing out more applications of the coresets. They should have further applications in the design of distributed or streaming algorithms.

**Strengths:**

The paper gives a new approach to a natural problem. Coresets construction should be known

They can achieve speed up practically using their ideas

**Weaknesses:**

The algorithmic and analysis ideas are simple

**Questions:**

Have the authors thought of applying these ideas to MPC for parallel computing algorithms?  I think they should immediately lead to results in such settings.

**Limitations:**

yes

---

> ### Author Rebuttal · Authors · 2023-08-05
>
> We thank you for reviewing our paper.
>
> We emphasize that our core-set results **do imply** streaming and distributed algorithms for the considered problem. As mentioned in our introduction [in the paragraph right before Section 1.1], the application of core-sets to streaming and distributed models is explained in detail in the reference [11] (IMMM, 2014).
>
> Let us describe how our core-set result implies an algorithm in the MPC model:
>
> - First suppose that the data of each group, i.e., $P_i$, is given to the $i$-th machine, each machine locally computes a core-set of small size (e.g. $O(k^2)$) on its own data and sends it to a single aggregator machine. The aggregator then runs some offline algorithm and computes the solution. Because of the composability property of the core-sets, the solution computed by the aggregator is a good approximation for the union of the whole data, i.e., P.
> Thus, we get an MPC algorithm whose communication is only $O(k^2)$ per machine ($m k^2$ in total, where $m$ is the number of groups) and the algorithm runs in a single round.
>
> - Even if the partitioning of the input data is done arbitrarily as opposed to being based on the groups, a slight modification of the above gives an MPC algorithm. In particular, each machine now computes a core-set of size $mk^2$ as follows: For each of its subset of points from each group it computes a core-set of size $k^2$ and takes their union. Thus the overall communication of all machines is $Mmk^2$ (where M is the number of machines and $m$ is the number of groups). Again the algorithm runs in one round.
>
> We will include this application to the MPC model in the paper.

---

### Author Rebuttal · Authors · 2023-08-05

We sincerely thank all four reviewers for the positive feedback and their valuable comments. We provided a response individually to all the reviewers. Please let us know if there are additional questions that we can address during the discussion phase.

---

### Decision · Program_Chairs · 2023-09-21

**Decision:**

Accept (poster)

**Comment:**

Overall, reviewers were positive about the paper. The model for data summarization is natural. The main shortcoming is that the techniques are not particularly deep. Still, the results will likely be of interest.